# LabelDP-Pro: Learning with Label Differential Privacy via Projections

**Badih Ghazi**
Google Research

**Yangsibo Huang**[*]
Princeton University

**Pritish Kamath**
Google Research

**Ravi Kumar**
Google Research

**Pasin Manurangsi**
Google Research

**Chiyuan Zhang**
Google Research

## ABSTRACT

Label differentially private (label DP) algorithms seek to preserve the privacy of the labels in a training dataset in settings where the features are known to the adversary. In this work, we study a new family of label DP training algorithms. Unlike most prior label DP algorithms that have been based on label randomization, our algorithm naturally leverages the power of the central model of DP. It interleaves gradient projection operations with private stochastic gradient descent steps in order to improve the utility of the trained model while guaranteeing the privacy of the labels. We show that such projection-based algorithms can be made practical and that they improve on the state-of-the-art for label DP training in the high-privacy regime. We complement our empirical evaluation with theoretical results shedding light on the efficacy of our method through the lens of bias-variance trade-offs.

## 1 INTRODUCTION

Differential privacy (DP) (Dwork et al., 2006b;a), which introduces calibrated noise into machine learning pipelines, has emerged as a fundamental tool for safeguarding the privacy of user data. This formal and robust framework, governed by a *privacy parameter* $\varepsilon$ (see Definition 1), is widely used in private machine learning applications.

In machine learning, the vanilla definition of DP aims to protect both the features and labels of the

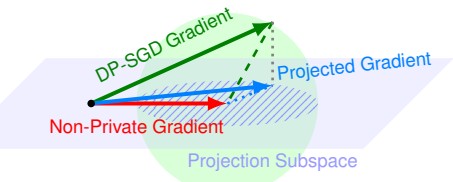

Figure 1: LabelDP-Pro denoises the DP-SGD gradient via projection. Since the signal (non-private gradient) is already in the subspace, the projection essentially reduces the additive DP-SGD noise. The green dashed (resp., blue dotted) line shows the gradient noise before (resp., after) projection.

training examples. However, this can be an overkill in settings where the sensitivity lies solely in the labels of the training examples. A notable example is in the realm of online advertising (Cheng et al., 2016; Wang et al., 2017; Guo et al., 2017; Zhou et al., 2018; Naumov et al., 2019; Wang et al., 2021). Here, the primary objective is to predict whether a user will perform a useful action on the advertiser website (e.g., purchase the advertised product) after interacting with an advertisement on another (the publisher's) website, a task inherently involving *private* and sensitive labels. This prediction is made based on features consisting of the product description associated with the displayed advertisement, which are generally considered *non-private*. [1]

The notion of label differential privacy (LabelDP), introduced by Chaudhuri & Hsu (2011), captures such use cases by leveraging the standard definition of DP to only protect the privacy of the labels. There has been a growing interest in studying label-only privacy in deep learning (e.g., Ghazi et al. (2021); Malek et al. (2021); Esfandiari et al. (2022); Tang et al. (2022)). Many of these

---

[*]Work done while interning at Google Research.

[1]We point out that several new privacy-preserving APIs for ad measurement that were proposed by different web browsers support training ad prediction models with label DP. These include APIs from Mozilla and Meta (Thomson, 2022), Google Chrome (Nalpas & White, 2021), and Apple Safari (app).

mechanisms rely on variants of the classic randomized response algorithm (Warner, 1965). In such mechanisms, each label is randomly changed to a (potentially) different label according to a pre-defined probability distribution. The training is then conducted with the randomized labels, which ensures LabelDP.

In this paper, we focus on the *high-privacy* regime (with a small privacy parameter $\varepsilon$). This regime is especially important where highly sensitive data are being processed, as the privacy guarantee becomes exponentially loose with large $\varepsilon$. Additionally, small $\varepsilon$ is also essential for *user-level* privacy where the privacy budget is shared by multiple examples of each user. Therefore, small $\varepsilon$ is generally desired in private learning when possible. However, achieving LabelDP in the high-privacy regime presents a unique challenge. The crux of the issue lies in the fact that in the high-privacy regime, the signal-to-noise ratio is extremely low in the privatized labels generated by randomized response. This problem is encountered by all LabelDP mechanisms that are based on randomized response and its variants.

On the other hand, we notice that DP-SGD (Abadi et al., 2016), a standard method for training deep networks with DP guarantees, exhibits better noise scaling in the high-privacy regime, compared to mechanisms relying on randomized response. However, DP-SGD guarantees privacy for both the features and the labels, which is overly stringent for the LabelDP setting. Therefore, to fully harness the advantage of DP-SGD's noise scaling, we study LabelDP-Pro, a family of methods that tailor DP-SGD to unlock better utility in the high-privacy regime by applying projection operations.

Our main contributions are summarized as follows:

- We present LabelDP-Pro to adapt DP-SGD for LabelDP settings with a projection-based Denoiser. This Denoiser leverages the public input features in LabelDP to counteract the over-killing noise associated with full (i.e., features and labels) DP protection in DP-SGD (Section 3).
- We show an efficient method for computing the projections via advanced autodiff (Section 3.2).
- Our LabelDP-Pro is the first practical method that fully utilizes the flexibility of the LabelDP definition, the central DP setting, and the approximate-DP guarantee. In contrast, all LabelDP mechanisms based on randomized response operate in the local DP setting and are restricted to offer pure-DP guarantees, resulting in poor utility in high-privacy regimes.
- Through empirical evaluations on four benchmark datasets, we show that LabelDP-Pro improves the state-of-the-art for LabelDP training in the high-privacy regime (Section 5).
- We extend our investigation to user-level DP, which ensures the privacy of *all* examples contributed by each user, and demonstrate using a real-world dataset for which LabelDP-Pro offers a consistent improvement over previous LabelDP baselines (Section 6).
- Complementing our empirical findings, we also provide theoretical analyses that justify the choice of the Denoiser and explain the observed improvements (Section 4).

## 2 LEARNING WITH LABEL DIFFERENTIAL PRIVACY

Let $\mathcal{D}$ be an unknown distribution on $\mathcal{X} \times \mathcal{Y}$. We consider the multi-class classification setting where $\mathcal{Y} = [K] := \{1, \ldots, K\}$. In supervised learning, we have a training set $D$ of $n$ examples drawn i.i.d. from $\mathcal{D}$. The goal is to learn a predictor $f_\theta : \mathcal{X} \to \mathbb{R}^K$ to minimize the expected loss $\mathcal{L}_\mathcal{D}(f_\theta) := \mathbb{E}_{(x,y)\sim\mathcal{D}}\ell(f_\theta(x), y)$, for some loss function $\ell : \mathbb{R}^K \times \mathcal{Y} \to \mathbb{R}$; the most common loss function for multi-class classification is the *cross entropy loss* $\ell(v, y) = -\log v_y$. With an abuse of notation, we also use $\ell(\theta, (x, y))$ to denote the per-example loss $\ell(f_\theta(x), y)$.

### 2.1 (LABEL) DIFFERENTIAL PRIVACY

We recall the definition of DP; for more background, see the book of Dwork & Roth (2014b).

**Definition 1** (DP; Dwork et al. (2006b)). *Let $\varepsilon \in \mathbb{R}_{>0}, \delta \in [0, 1)$. A randomized algorithm $\mathcal{A}$ is said to be $(\varepsilon, \delta)$-differentially private (denoted $(\varepsilon, \delta)$-DP) if for any two adjacent input datasets (differ by at most one datapoint) $D$ and $D'$, and any subset $S$ of outputs of $\mathcal{A}$, it holds that $\Pr[\mathcal{A}(D) \in S] \leq e^\varepsilon \cdot \Pr[\mathcal{A}(D') \in S] + \delta$.*

In supervised learning, a learning algorithm produces a model as its output, while the labeled training set serves as its input. Two training datasets are "adjacent" if they differ on a single training example. This notion of adjacency protects the privacy of both the features and the label of any individual

---

**Algorithm 1:** LabelDP-Pro

---

**Input** : Initial model weights $\theta_0$, a training set $D$ of size $n$, learning rate $\eta_t$, number of training iterations $T$, batch size $n_1$, noise multiplier $\sigma$, gradient norm bound $C$, and Denoiser, which is a method for denoising gradients.

**Output:** The trained model weights $\theta_T$.

1 **for** $t = 0$ **to** $T - 1$ **do**
2   Get a mini-batch $I_t^G$ of $n_1$ indices by uniform sampling from $[n]$
3   **foreach** $i \in I_t^G$ **do** $\mathbf{g}_t(x_i, y_i) \leftarrow \nabla_{\theta_t} \ell(\theta_t, (x_i, y_i))$     `// Compute gradient`
4   **foreach** $i \in I_t^G$ **do** $\bar{\mathbf{g}}_t(x_i, y_i) \leftarrow \mathbf{g}_t(x_i, y_i) / \max\left(1, \frac{\|\mathbf{g}_t(x_i, y_i)\|_2}{C}\right)$    `// Clip gradient`
5   $\tilde{\mathbf{g}}_t \leftarrow \frac{1}{n_1}\left(\sum_{i \in I_t^G} \bar{\mathbf{g}}_t(x_i, y_i) + \mathcal{N}(0, \sigma^2 C^2 \mathbf{I})\right)$    `// Aggregate and add noise`
6   $\hat{\mathbf{g}}_t \leftarrow \mathsf{Denoiser}\left(\tilde{\mathbf{g}}_t, \theta_t, \{x_i\}_{i \in I_t^G}\right)$     `// De-noise the gradient`
7   $\theta_{t+1} \leftarrow \theta_t - \eta_t \hat{\mathbf{g}}_t$     `// Optimize; can also use other optimizers`

---

example. However, in certain scenarios, protecting the features may be unnecessary or infeasible, and the focus is solely on protecting the privacy of the labels. This leads to the following notion:

**Definition 2** (LabelDP; Chaudhuri & Hsu (2011)). *A randomized training algorithm $\mathcal{A}$ is $(\varepsilon, \delta)$-LabelDP if it is $(\varepsilon, \delta)$-DP when two input datasets differ on the label of a single training example.*

For both notions, due to the $e^\varepsilon$ factor, the *high-privacy regime* corresponds to small $\varepsilon$ (e.g., $\varepsilon < 1$).

**Item-level and user-level DP** In the traditional notion of DP, the goal is to protect the privacy of each training example, where it is assumed that each user contributed only one training example; this is commonly known as *item-level* or *example-level* privacy. However, in scenarios where a single user contributes multiple training examples, a more desirable objective would be the so-called *user-level* privacy, which ensures the privacy of all examples contributed by one user. In this case, the "adjacent" dataset would differ on all the examples from a single user. This is especially relevant in applications such as online advertising and federated learning, where each user often contributes multiple examples (McMahan et al., 2017; Liu et al., 2020; Levy et al., 2021).

## 2.2 RANDOMIZED RESPONSE AND DP-SGD

Randomized response (RR) (Warner, 1965), which precedes the advent of DP, has been used as a standard mechanism to provide LabelDP guarantees. Let $y \in [K]$ be the ground truth label. When an observer queries the value of $y$, RR with $\varepsilon$-LabelDP responds with a random draw $\tilde{y} \in [K]$. Specifically, RR returns the ground truth label $y$ with a probability of $\frac{e^\varepsilon}{e^\varepsilon + K - 1}$, while with the remaining probability, it returns a uniformly random element from $[K] \setminus \{y\}$.

DP-SGD was proposed in the seminal paper by Abadi et al. (2016), and has become the most widely adopted algorithm for privately training deep neural networks. DP-SGD works by clipping the per-example gradient and adding Gaussian noises, during each training iteration. (For a formal description, see Algorithm 1 with a NOOP denoiser.) DP-SGD relies on the fact that only a random mini-batch of examples are processed in each iteration, and uses *amplification by sampling* (Abadi et al., 2016; Bassily et al., 2014; Wang et al., 2018) to drive down the overall privacy cost.

## 3 LabelDP-Pro

As discussed above, RR-based LabelDP mechanisms flip the labels with probability $\frac{K-1}{e^\varepsilon + K - 1}$; thus, in the high-privacy regime, incur exponentially more noise. For instance, with $K = 10$ and $\varepsilon = 0.5$, in expectation more than 84% of the training labels would be incorrect. In contrast, the scale of the Gaussian noise in DP-SGD increases only linearly with $1/\varepsilon$ in the high-privacy regime. The suboptimal noise scaling for RR is partly because it provides a more stringent *local* privacy guarantee, compared to the *central* privacy guarantee discussed here (See Appendix B for more details). Inspired by this, we propose LabelDP-Pro, a novel approach that enjoys the flexibility of both central privacy (improved noise scaling) and the LabelDP (auxiliary information computed from the input features) to unlock high utility of LabelDP in the high-privacy (i.e., small $\varepsilon$) regime.

Table 1: A summary of denoisers for noisy gradients $\tilde{\mathbf{g}}_t$, with optional access to the model weights $\theta_t$ and the features of the current batch $:\{x_i\}_{i \in I_t^G}$. A denoiser can also access the features of the full training set. In particular, it can sample an "alternative" batch $I_t^P$ of $n_2$ examples and have access to $\{x_i\}_{i \in I_t^P}$. The last column indicates if the denoiser is compatible with amplification by subsampling in the privacy accounting.

| Denoiser | Output | Compatible with Amp. |
|---|---|---|
| NoOp | $\tilde{\mathbf{g}}_t$ | Yes |
| SelfSpan | $\text{Proj}_A(\tilde{\mathbf{g}}_t)$ for $A = \text{span}\left(\{\nabla_{\theta_t}\ell(\theta_t, (x_i, \kappa))\}_{i \in I_t^G, \kappa \in [K]}\right)$ | No |
| SelfConv | $\text{Proj}_A(\tilde{\mathbf{g}}_t)$ for $A = \text{conv}\left(\{\nabla_{\theta_t}\ell(\theta_t, (x_i, \kappa))\}_{i \in I_t^G, \kappa \in [K]}\right)$ | No |
| AltConv | $\text{Proj}_A(\tilde{\mathbf{g}}_t)$ for $A = \text{conv}\left(\{\nabla_{\theta_t}\ell(\theta_t, (x_i, \kappa))\}_{i \in I_t^P, \kappa \in [K]}\right)$ | Yes |

## 3.1 PROJECTION-BASED GRADIENT DENOISING

Due to better noise scaling, DP-SGD could achieve better utility than classical algorithms like RR in the high-privacy LabelDP regime. However, it would be an overkill as DP-SGD protects the privacy of both features and labels. In the following, we propose a framework to improve DP-SGD by using the input features, which yields a family of new algorithms that enjoys the same linear noise scaling, and outperforms DP-SGD (and RR) in the high-privacy LabelDP regime.

The main idea is that structural priors can be computed based on the input features, and utilized to *denoise* the noisy gradient from DP-SGD. Specifically, the framework introduces a Denoiser function, described in Algorithm 1 (vanilla DP-SGD is thus a special case with a NoOp denoiser.). Table 1 summarizes some useful denoisers. For example, the SelfSpan denoiser makes use of the fact that the (original unnoised) gradient lies in the span of the per-example per-class gradients $\{\nabla_{\theta_t}\ell(\theta_t, (x_i, \kappa))\}_{i \in I_t^G, \kappa \in [K]}$. So it denoises the private gradient from DP-SGD by projecting it onto this span. Note the span can be constructed using only the features and without accessing the labels—the denoising step thus crucially exploits the LabelDP setting! As an alternative view, note that the privatized gradient in DP-SGD is the sum of the signal (the gradient) and the Gaussian noise. The SelfSpan projection for the signal part is a no-op because the signal is already in the span. As a result, the denoising step is essentially reducing the noise by projecting it onto a lower dimensional subspace while keeping the signal intact (Figure 1).

Taking one step further, note that the unnoised gradient is not just in the span, but also in the convex hull of the per-example per-class gradients. This observation leads to the SelfConv denoiser, which has an improved dependency on $K$ (Table 4). Moreover, we can also independently sample an "alternative" batch $I_t^P$ of $n_2$ examples and project onto its gradients $\{\nabla_{\theta_t}\ell(\theta_t, (x_i, \kappa))\}_{i \in I_t^P, \kappa \in [K]}$, again using only the public information; we call this denoiser AltConv. Empirically, we found that both SelfConv and AltConv behave quite similarly under the same DP-SGD noise (see Table 3), but the latter has the extra benefit of being amenable to *amplification by subsampling*, resulting in lower privacy cost. We defer to Section 3.4 for the details on the privacy accounting. Unless otherwise specified, we will use the AltConv denoiser in the experiments.

## 3.2 MEMORY-EFFICIENT PROJECTION VIA ADVANCED AUTODIFF

The denoisers (Table 1) operate by projecting the noisy gradient onto the subspace or the convex hull spanned by a set of per-example per-class gradients. A naive implementation of such projection would need $n_2 K d$ memory to store the vertices spanning the convex hull, where $d$ is the model size.

We next describe a memory-efficient algorithm. Let $\tilde{\mathbf{g}}$ be the privatized gradient, and $\mathbf{G} = [\nabla_\theta \ell(\theta, (x_i, \kappa))]_{i \in I^P, \kappa \in [K]}$ be the $d \times n_2 K$ dimensional matrix of the per-example per-class gradients (in the case of SelfSpan and SelfConv, $I^P = I^G$ and $n_2 = n_1$). To project $\tilde{\mathbf{g}}$ onto the convex hull of the columns of $\mathbf{G}$, we solve $\min_{\boldsymbol{\alpha} \in \Delta} \|\mathbf{G}\boldsymbol{\alpha} - \tilde{\mathbf{g}}\|^2$, where $\Delta$ is the unit simplex for $n_2 K$-dimensional vectors. We solve this optimization problem with projective gradient descent. Specifically, let $\boldsymbol{\alpha}_0 \in \Delta$ be an initialization of the projection coefficients, and $\eta^{\text{PGD}}$ be the step size, we iteratively update it as $\boldsymbol{\alpha}_{t+1} \leftarrow \text{Proj}_\Delta\left(\boldsymbol{\alpha}_t - 2\eta^{\text{PGD}} \boxed{\mathbf{G}^\top(\mathbf{G}\boldsymbol{\alpha}_t - \tilde{\mathbf{g}})}\right)$. The projection onto the unit simplex can be computed using the methods of Wang & Carreira-Perpiñán (2013). We next focus on efficiently computing the $\boxed{\text{boxed red}}$ part, which depends on two subroutines: $\mathbf{u} \mapsto \mathbf{G}\mathbf{u}$ for any $\mathbf{u} \in \mathbb{R}^{n_2 K}$, and $\mathbf{v} \mapsto \mathbf{G}^\top \mathbf{v}$ for any $\mathbf{v} \in \mathbb{R}^d$. We note that given the special structure of $\mathbf{G}$, those subroutines can be computed using advanced *auto differentiation* (autodiff, Baydin et al., 2018) primitives without materializing $\mathbf{G}$. Indeed, define $\mathbf{L} : \theta \mapsto [\ell(\theta, (x_i, \kappa))]_{i \in [n_2], \kappa \in [K]}$ and observe

that $\mathbf{G}^\top$ is the Jacobian of $\mathbf{L}$. Then $\mathbf{G}^\top \mathbf{v}$ is the *Jacobian-vector product* (JVP). It turns out that JVP can be implemented via the *forward mode autodiff*, which cumulatively computes the results in a "forward" manner from the bottom of the neural network, and does *not* require a materialization of $\mathbf{G}$. Similarly, the *reverse mode autodiff* can be used to compute the *vector-Jacobian product* (VJP) in $\mathbf{Gu} = (\mathbf{u}^\top \mathbf{G}^\top)^\top$; again, VJP does *not* need to materialize $\mathbf{G}$. In fact, the reverse mode autodiff is the default algorithm for computing the gradients in most deep learning frameworks, since the *back-propagation* algorithm (Rumelhart et al., 1986) is a special case of it with a scalar loss. We give a pseudocode implementation in Appendix C.

## 3.3 STABLE PROJECTION VIA COEFFICIENT SMOOTHING

Due to the noise in the privatized gradient, the projection coefficients $\boldsymbol{\alpha}$ sometimes put dominating values on an "incorrect" class. This might cause numerical instabilities in the learning dynamics due to excessively large gradient norms (Figure 2) because of the unboundedness of the cross-entropy loss, as also observed in previous work (Ghosh et al., 2017; Ma et al., 2020; Zhou et al., 2021).

In our case, we found a simple smoothing-based regularization to be effective. Specifically, we create a smoothed projection coefficient $\tilde{\boldsymbol{\alpha}} = \lambda\boldsymbol{\alpha} + (1-\lambda)\boldsymbol{\beta}$, where $\boldsymbol{\beta} \in \mathbb{R}^{n_2 K}$ is a constant vector with each coordinate being $\frac{1}{K \times n_2}$, representing a uniform distribution that assigns equal weights to all projection vertices. $\lambda$ controls the level of regularization, with smaller values indicating stronger regularization. As depicted in Figure 2 and summarized in Table 2, enabling regularization ($\lambda < 1$) leads to a more stable training process and improves utility. Typically, we find that a $\lambda$ value of $0.75$ yields the best results. Note if $\boldsymbol{\alpha}$ is a one-hot encoded multi-class label, then this regularization is akin to the label smoothing technique (Müller et al., 2019; Lukasik et al., 2020).

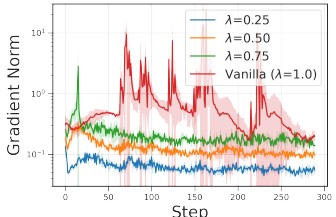 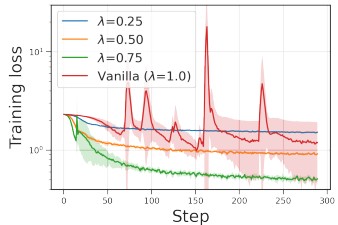

Figure 2: Effect of the regularization coefficient $\lambda$ on the gradient norm (left) and the training loss (right). Applying regularization with $\lambda < 1$ helps stabilize the training.

Table 2: MNIST accuracy (%) for the implementation with regularization compared to the vanilla implementation.

| $\varepsilon$ | Vanilla | w/ Regularization |
|---|---|---|
| 0.1 | 86.2 | **91.1** |
| 0.2 | 87.9 | **92.9** |
| 0.5 | 88.2 | **94.1** |
| 1.0 | 91.3 | **94.3** |
| 2.0 | 92.2 | **94.5** |

## 3.4 PRIVACY ANALYSIS

We recall the privacy analysis of DP-SGD (i.e., LabelDP-Pro with NOOP denoiser), and then show that the same analysis also holds for LabelDP-Pro with ALTCONV denoiser. The privacy analysis of DP-SGD is usually done by considering an approximation, where instead of shuffling and batch-partitioning as generally done in practice, in each iteration, a batch is constructed by sampling each example independently with probability $n_1/n$. This enables analyzing the privacy cost of DP-SGD via the composition of adaptively chosen Poisson subsampled Gaussian mechanisms, which uses privacy amplification by subsampling to provide lower privacy cost. It is now easy to see that applying the same approximation to LabelDP-Pro using ALTCONV denoiser is simply a composition of adaptively chosen Poisson subsampled Gaussian mechanisms *with post-processing*, hence we can apply the post-processing property of DP (Dwork & Roth, 2014a, Proposition 2.1).

The main thing to note about the ALTCONV denoiser is that the projection step involves sampling a random subset $I_t^P$ from the training data to define the convex hull that the noisy gradient is projected to. The analysis goes through since these are independent of the examples in the batch $I_t^G$ and do not use the information about labels of examples in $I_t^P$. Note that the same analysis *does not hold* when using SELFSPAN or SELFCONV denoisers as the convex hull is defined by (unlabeled) examples in $I_t^G$, and so the final projected gradient might reveal information about $I_t^G$ (which examples ended up being sampled in the current batch) and breaks privacy amplification by subsampling.

Rényi Differential Privacy (RDP) accounting (Mironov, 2017) has been the most widely used approach in DP-SGD (Abadi et al., 2016). Numerical approaches for obtaining tighter estimates of privacy parameters have been studied recently using the notion of *privacy loss distributions* (PLD) (Meiser & Mohammadi, 2018; Sommer et al., 2019), and several algorithms have been

Table 3: Test accuracy (%) on the MNIST dataset with different denoisers. "AMP." stands for privacy amplification via subsampling. The SELFSPAN and SELFCONV denoiser should not use amplification, and the corresponding columns are shown (in gray) only for reference purposes.

| $\varepsilon$ | DP-SGD | Project onto span | | Project onto convex hull | | |
|---|---|---|---|---|---|---|
| | NoOp w/ AMP. | SELFSPAN w/o AMP. | SELFSPAN w/ AMP. | ALTCONV w/ AMP. | SELFCONV w/o AMP. | SELFCONV w/ AMP. |
| **0.1** | 85.6 | 64.5 | 86.1 | 91.1 | 65.9 | 91.3 |
| **0.2** | 87.9 | 73.8 | 88.2 | 92.9 | 75.6 | 93.3 |
| **0.5** | 90.9 | 81.8 | 91.5 | 94.1 | 83.8 | 94.2 |
| **1.0** | 92.8 | 84.8 | 92.9 | 94.3 | 86.1 | 94.3 |
| **2.0** | 93.8 | 87.4 | 93.6 | 95.4 | 88.2 | 95.6 |

Table 4: Additional excess error *due to privacy* for PGD using stochastic gradient oracles.

| Method | Additional excess error due to privacy |
|---|---|
| LabelDP-Pro: NOOP | $O\left(\frac{RC}{\sqrt{T}} \cdot \frac{\sqrt{d}\sigma_{\mathrm{a}}}{\sqrt{n_1}}\right)$ |
| LabelDP-Pro: SELFSPAN | $O\left(\frac{RC}{\sqrt{T}} \cdot \sqrt{K}\sigma_{\mathrm{na}}\right)$ |
| LabelDP-Pro: SELFCONV | $O\left(RC\left(\frac{1}{\sqrt{n_1}} + \frac{\sqrt{\sigma_{\mathrm{na}}}\sqrt[4]{\log(Kn_1)}}{\sqrt[4]{n_1}}\right)\right)$ |
| LabelDP-Pro: ALTCONV | $O\left(RC\left(\frac{1}{\sqrt{n_1}} + \frac{1}{\sqrt{n_2}} + \frac{\sqrt{\sigma_{\mathrm{a}}}\sqrt[4]{\log(Kn_2)}}{\sqrt[4]{n_1}}\right)\right)$ |
| RR-Debiased ($p = \frac{K}{e^\varepsilon + K - 1}$) | $O\left(\frac{RC}{\sqrt{T}} \cdot \frac{\sqrt{p}}{(1-p)\sqrt{n_1}}\right)$ |

studied for the same (Koskela et al., 2020; Gopi et al., 2021; Ghazi et al., 2022; Doroshenko et al., 2022). In particular, we use the privacy accounting implementation in Google's DP Library (2020).

## 4 BOUNDS FOR CONVEX EMPIRICAL RISK MINIMIZATION

We now present our theoretical motivation that underlies the method presented in Algorithm 1 and the denoisers presented in Table 1. We examine this through a case study of stochastic convex optimization. Consider the goal of minimizing a *convex* objective $\mathcal{L}(\boldsymbol{w})$ for $\boldsymbol{w} \in \mathcal{K}$, for some convex set $\mathcal{K}$ with diameter $R$ using a stochastic gradient oracle that, given $\boldsymbol{w}$, returns a stochastic estimate $\tilde{g}(\boldsymbol{w})$ of $g(\boldsymbol{w}) := \nabla\mathcal{L}(\boldsymbol{w})$. The loss is said to be $C$-Lipschitz if $\|g(\boldsymbol{w})\|_2 \leq C$ for all $\boldsymbol{w}$. We say that the gradient oracle has bias $\alpha$ and variance $\omega^2$ if $\tilde{g}(\boldsymbol{w}) = g(\boldsymbol{w}) + e(\boldsymbol{w})$ such that $\|\mathbb{E}\,e(\boldsymbol{w})\|_2 \leq \alpha$ and $\mathbb{E}\,\|e(\boldsymbol{w}) - \mathbb{E}\,e(\boldsymbol{w})\|_2^2 \leq \omega^2$ holds for all $\boldsymbol{w}$. Additionally, we say that the gradient oracle has expected squared norm of $\widetilde{C}^2$ if $\mathbb{E}\,\|\tilde{g}(\boldsymbol{w})\|^2 \leq \widetilde{C}^2$ for all $\boldsymbol{w}$. The *projected gradient descent* (PGD) with step size $\eta$ is defined as iteratively performing $\boldsymbol{w}_{t+1} \leftarrow \mathsf{Proj}_{\mathcal{K}}(\boldsymbol{w}_t - \eta g(\boldsymbol{w}_t))$, where $\mathsf{Proj}_{\mathcal{K}}(\cdot)$ is the projection onto $\mathcal{K}$. Let $\boldsymbol{w}^*$ be a minimizer of $\mathcal{L}(\boldsymbol{w})$. The expected excess loss can be bounded as shown below following the standard analysis of PGD (see, e.g., Hazan (2022)); we include a proof for completeness in Appendix E.

**Lemma 3.** *For a convex loss function $\mathcal{L}$, and a gradient oracle with bias $\alpha$ and expected squared norm $\widetilde{C}^2$, PGD over a convex set $\mathcal{K} \subseteq \mathbb{R}^d$ with diameter $R$ and step size $\eta = \frac{R}{\widetilde{C}\sqrt{T}}$ achieves the following expected excess risk bound for the average iterate $\bar{\boldsymbol{w}}_T := \frac{1}{T}\sum_{t=0}^{T-1} \boldsymbol{w}_t$:*

$$\mathbb{E}\left[\mathcal{L}\left(\bar{\boldsymbol{w}}_T\right)\right] - \mathcal{L}(\boldsymbol{w}^*) \ \leq \ \frac{R\widetilde{C}}{\sqrt{T}} + \alpha R. \tag{1}$$

Note that if $\mathcal{L}$ is $C$-Lipschitz and the gradient oracle has variance $\omega^2$, then $\widetilde{C}^2 \leq (C + \alpha)^2 + \omega^2$, although it is possible to get an improved upper bound on $\widetilde{C}$ for some oracles we consider.

Now consider the problem of minimizing the *empirical loss*. Namely, let $\ell : \mathbb{R}^d \times \mathcal{X} \times [K] \to \mathbb{R}$ be a $C$-Lipschitz loss function, and let $\mathcal{L}(\boldsymbol{w}) := \frac{1}{n}\sum_{i\in[n]} \ell(\boldsymbol{w}; x_i, y_i)$. Observe that the true gradient at $\boldsymbol{w}$ is given as $\boldsymbol{g}(\boldsymbol{w}) = \frac{1}{n}\sum_{i\in[n]} \nabla_{\boldsymbol{w}}\ell(\boldsymbol{w}; x_i, y_i)$, and each method discussed in this paper provides a stochastic gradient oracle of the empirical loss $\mathcal{L}$. In particular, we discuss LabelDP-Pro using various denoisers as well as the debiased randomized response for this theoretical case study.

We can apply Lemma 3 to bound the excess error of PGD using the stochastic gradient oracles arising out of each method. Table 4 provides upper bounds on the excess error for each method; we derive these formally in Appendix E. The most technical part is to bound the expected squared norm of the error in the stochastic gradient oracle with the ALTCONV denoiser, as shown below, which in particular implies a bound on both the bias and the variance of the oracle.

**Lemma 4.** *Let $I^G, I^P \subseteq [n]$ be random subsets of size $n_1, n_2$ respectively. Let $\tilde{g}$ denote the noisy gradient estimate from $I^G$, namely, $\tilde{g} := \frac{1}{n_1} \sum_{i \in I^G} \nabla_{\boldsymbol{w}} \ell(\boldsymbol{w}; x_i, y_i) + \boldsymbol{e}$ for $\boldsymbol{e} \sim \mathcal{N}(0, \frac{\sigma^2 C^2}{n_1} \cdot \mathbf{I})$. Let $A = \mathsf{conv}\left(\{\nabla_{\boldsymbol{w}} \ell(\boldsymbol{w}, (x_i, \kappa))\}_{i \in I^P, \kappa \in [K]}\right)$ denote the convex body on which we project. Then, over the choice of $I^G, I^P$, and $\boldsymbol{e}$, we have*

$$\mathbb{E}\left[\|\operatorname{Proj}_A(\tilde{g}) - g\|_2^2\right] \leq O\left(C^2 \cdot \left(\frac{1}{n_1} + \frac{1}{n_2} + \frac{\sigma\sqrt{\log(Kn_2)}}{\sqrt{n_1}}\right)\right).$$

We now interpret the bounds in Table 4, and compare it against the accuracy bounds in Table 3. The rate of SELFSPAN gets rid of the dependence on dimension $d$; however, it comes at a price that privacy amplification by subsampling is no longer applicable, due to which the noise multiplier $\sigma_{\mathrm{na}}$ is larger than $\sigma_{\mathrm{a}}$. This is indeed limiting, and if we instead use $\sigma_{\mathrm{a}}$ instead, SELFSPAN denoiser ends up better than DP-SGD. The rate of SELFCONV improves upon the rate of SELFSPAN, but again suffers from the same limitation that privacy amplification by subsampling is not applicable. Finally, ALTCONV achieves error that is slightly larger than SELFCONV when using the same noise multiplier (which can be see by comparing against SELFCONV when we use the noise multiplier using privacy amplification), but since ALTCONV benefits from privacy amplification by subsampling, it overall ends up being better than SELFCONV.

Note that the bounds in Table 4 hold only for convex loss functions. Moreover, they are upper bounds on the excess error. For instance, while the bound in Lemma 3 is tight in the worst case, better rates might be achievable under additional assumptions on the loss function. Despite these limitations, we see in Table 4 that the excess error bounds align well with the experimental findings.

## 5 EVALUATION WITH ITEM-LEVEL PRIVACY

In this section, we evaluate the performance of LabelDP-Pro and compare it with baseline approaches for item-level privacy. We detail the experimental setup in Section 5.1 and present the main results in Section 5.2. We then demonstrate that LabelDP-Pro can be combined with self-supervised learning to unlock even higher utility (Section 5.3).

### 5.1 EXPERIMENTAL SETUP

We compare our LabelDP-Pro on four benchmark datasets commonly used in LabelDP learning evaluations: MNIST (LeCun et al., 1998), kMNIST (Clanuwat et al., 2018), FashionMNIST (Xiao et al., 2017), and CIFAR-10 (Krizhevsky et al., 2009); against four established LabelDP baselines:

- Randomized Response (**RR**) (Warner, 1965): Method that randomly flips the label ( Section 2.2).
- RR with Debiased Loss (**RR-Debiased**): RR extension with unbiased loss function for training.
- Label Privacy Multi-Stage Training (**LP-MST**) (Ghazi et al., 2021): LP-MST trains the model on disjoint partitions of the dataset in different stages. Specifically, we compare against LP-2ST, a specific configuration with two training stages.
- Additive Laplace with Iterative Bayesian Inference (**ALIBI**) (Malek et al., 2021): A soft RR mechanism that directly adds Laplace noises to one-hot encodings of training labels.

We also report the performance for DP-SGD as a reference, since our approach is built upon it. See Appendix D for more details on the experiments.

### 5.2 MAIN RESULTS

As presented in Table 5, across both the MNIST and CIFAR-10 benchmarks, LabelDP-Pro consistently surpasses baseline LabelDP mechanisms when $\varepsilon < 1.0$. Notably, the performance gap widens as $\varepsilon$ decreases. For instance, when $\varepsilon = 0.2$, previous baselines all achieve accuracy levels close to random guessing, while LabelDP-Pro maintains non-trivial accuracy at 92.9% on MNIST and 30.8% on CIFAR-10. These results demonstrate the effectiveness of LabelDP-Pro in preserving utility, and they echo the theoretical analyses in Section 4. There also appears to be a threshold $\varepsilon^*$ such that when $\varepsilon \leq \varepsilon^*$, LabelDP-Pro outperforms RR, but when $\varepsilon > \varepsilon^*$, RR exhibit superior performance. In Appendix D.4, we show how the number of classes ($K$) and the probability for catastrophic failure ($\delta$) affect $\varepsilon^*$.

Compared to DP-SGD, LabelDP-Pro also consistently exhibits superior performance, underscoring the effectiveness of our Denoiser. Notably, this performance gap tends to be more pronounced when

Table 5: Test accuracy on (a) MNIST-10 and (b) and CIFAR-10, for different $\varepsilon$'s. The best result for each $\varepsilon$ is **boldfaced**. We report the utility differences between LabelDP-Pro and the best baseline among RR, RR-Debiased, LP-2ST, ALIBI ($\Delta_{\text{Baseline}}$), and between LabelDP-Pro and DP-SGD ($\Delta_{\text{DP-SGD}}$). LabelDP-Pro outperforms baseline methods in the high-privacy regime.

| Privacy Budget ($\varepsilon$) | Baseline LabelDP Methods | | | | DP-SGD | LabelDP-Pro (Ours) | | |
|---|---|---|---|---|---|---|---|---|
| | RR | RR-DEBIASED | LP-2ST | ALIBI | | Acc. | $\Delta_{\text{Baseline}}$ | $\Delta_{\text{DP-SGD}}$ |
| (a) MNIST-10 | | | | | | | | |
| 0.01 | 6.5 | 10.8 | 9.0 | 10.3 | 65.3 | **68.2** | 57.4 ↑ | 2.9 ↑ |
| 0.1 | 7.3 | 10.9 | 10.0 | 9.7 | 85.6 | **91.1** | 80.2 ↑ | 5.5 ↑ |
| 0.2 | 8.1 | 10.9 | 10.0 | 9.7 | 87.9 | **92.9** | 82.0 ↑ | 5.0 ↑ |
| 0.5 | 10.4 | 11.0 | 10.0 | 82.5 | 90.9 | **94.1** | 83.1 ↑ | 3.2 ↑ |
| 1.0 | 51.3 | 50.6 | 83.3 | 94.1 | 92.8 | **94.3** | 11.0 ↑ | 1.5 ↑ |
| 2.0 | 96.7 | **96.8** | **96.8** | 96.6 | 93.8 | 95.4 | 1.4 ↓ | 1.6 ↑ |
| (b) CIFAR-10 | | | | | | | | |
| 0.05 | 10.0 | 10.0 | 10.0 | 9.9 | 10.1 | **16.0** | 6.0 ↑ | 5.9 ↑ |
| 0.1 | 10.0 | 10.0 | 10.0 | 9.9 | 24.6 | **27.0** | 17.0 ↑ | 2.4 ↑ |
| 0.2 | 10.0 | 10.0 | 10.0 | 17.9 | 29.8 | **30.8** | 12.9 ↑ | 1.0 ↑ |
| 0.5 | 10.0 | 10.0 | 35.8 | 34.4 | 36.9 | **40.2** | 5.8 ↑ | 3.3 ↑ |
| 1.0 | 17.0 | 10.0 | **60.9** | 51.3 | 43.5 | 44.8 | 16.1 ↓ | 1.3 ↑ |

Table 6: Test accuracy (%) on CIFAR-10 using Wide-ResNet18 with SelfSL. The PATE-FM results (also using Wide-ResNet18 but with SemiSL) are from Table 1 of Malek et al. (2021). The best result for each $\varepsilon$ is **boldfaced**. We report the utility differences between LabelDP-Pro and the best baseline among RR, RR-Debiased, RRWithPrior, PATE-FM ($\Delta_{\text{Baseline}}$), and between LabelDP-Pro and DP-SGD ($\Delta_{\text{DP-SGD}}$). Despite PATE-FM's stronger SemiSL pipeline (95.5% non-private accuracy, compared to our 90.9%), LabelDP-Pro still outperforms PATE-FM in the high-privacy regime.

| $\varepsilon$ | Baseline LabelDP Methods | | | | DP-SGD | LabelDP-Pro (Ours) | | |
|---|---|---|---|---|---|---|---|---|
| | RR | RR-Debiased | RRWithPrior | PATE-FM | | Acc. | $\Delta_{\text{Baseline}}$ | $\Delta_{\text{DP-SGD}}$ |
| **0.01** | 2.5 | 2.7 | 24.6 | - | 71.8 | **74.4** | 43.1 ↑ | 2.6 ↑ |
| **0.02** | 3.4 | 2.1 | 50.4 | - | 81.6 | **82.4** | 32.0 ↑ | 0.8 ↑ |
| **0.05** | 3.5 | 2.7 | 70.8 | - | 85.1 | **85.5** | 14.7 ↑ | 0.4 ↑ |
| **0.18** | 4.7 | 3.1 | 76.8 | 73.4 | 86.4 | **86.6** | 9.8 ↑ | 0.2 ↑ |
| **0.29** | 5.1 | 3.7 | 76.8 | 86.9 | 87.0 | **87.2** | 0.3 ↑ | 0.2 ↑ |
| **0.5** | 6.0 | 9.8 | 78.1 | - | 87.1 | **87.4** | 9.3 ↑ | 0.3 ↑ |
| **1.6** | 88.1 | 88.0 | 83.1 | **93.7** | 87.2 | 87.7 | 6.5 ↓ | 0.5 ↑ |

dealing with smaller $\varepsilon$'s which can be potentially attributed to the fact that when $\varepsilon$ is relatively large, the noise introduced by DP-SGD is already minimal, leaving limited room for improvement by our Denoiser. We also observe similar results on other datasets (see Appendix D.4).

## 5.3 LEVERAGING SELF-SUPERVISED LEARNING WITH LabelDP

Since the features are public in LabelDP, we can use self-supervised learning (SelfSL) techniques (Chen et al., 2020a;b; Grill et al., 2020; He et al., 2020; Caron et al., 2021; Xie et al., 2022) to obtain high-quality representations to further the utility of LabelDP Ghazi et al. (2021). These representations can be learned solely from the input features without requiring labels; later, they can be fed into the private supervised training pipelines alongside the corresponding labels.

We demonstrate this approach by evaluating LabelDP-Pro with SelfSL on CIFAR-10. Specifically, we pre-train Wide-ResNet18 (He et al., 2016) as a representation extractor with unlabeled CIFAR-10 images using the SimCLR algorithm (Chen et al., 2020a). We then train classifiers on the extracted representations using various LabelDP mechanisms, including RR, RR-Debiased, RRWithPrior (Ghazi et al., 2021)[2], DP-SGD, and LabelDP-Pro. See Appendix D.2 for more details.

We additionally include PATE-FM (Malek et al., 2021), a state-of-the-art semi-supervised learning (SemiSL) based method, for comparison. It is noteworthy that PATE-FM benefits from a more powerful SemiSL pipeline, with a non-private accuracy of 95.5% compared to our SelfSL pipeline's

---

[2]RRWithPrior is a subroutine in the LP-MST method, where randomized labels are sampled with access to label priors to improve accuracy. When implementing this in the context of SelfSL, we follow the procedure detailed in Ghazi et al. (2021, Section 5.2): we cluster training examples based on the SelfSL model representations, and then query label histograms for each group using the discrete Laplace mechanism to obtain label priors. The budget spared for the histogram query can be found in Appendix D.

Table 7: Test AUC of RR, DP-SGD, and LabelDP-Pro on the Criteo data, with a different number of examples contributed per user ($k$) and privacy budget ($\varepsilon$). The best results for each ($\varepsilon, k$) combination are **boldfaced**.

| $\varepsilon$ | $k = 2$ | | | $k = 5$ | | | $k = 10$ | | |
|---|---|---|---|---|---|---|---|---|---|
| | RR | DP-SGD | LabelDP-Pro | RR | DP-SGD | LabelDP-Pro | RR | DP-SGD | LabelDP-Pro |
| **0.1** | 0.562 | 0.686 | **0.780** | 0.537 | 0.668 | **0.768** | 0.544 | 0.652 | **0.751** |
| **0.2** | 0.594 | 0.707 | **0.786** | 0.549 | 0.693 | **0.771** | 0.555 | 0.680 | **0.766** |
| **0.5** | 0.652 | 0.719 | **0.787** | 0.587 | 0.714 | **0.776** | 0.575 | 0.704 | **0.766** |
| **1.0** | 0.710 | 0.724 | **0.789** | 0.638 | 0.722 | **0.777** | 0.605 | 0.712 | **0.768** |
| **2.0** | 0.761 | 0.731 | **0.791** | 0.704 | 0.728 | **0.778** | 0.652 | 0.720 | **0.769** |
| **5.0** | **0.804** | 0.741 | 0.793 | 0.761 | 0.739 | **0.781** | 0.718 | 0.733 | **0.773** |
| $\infty$ | | 0.823 | | | 0.824 | | | 0.824 | |

non-private accuracy of $90.9\%$. As shown in Table 6, LabelDP-Pro consistently outperforms all previous LabelDP baselines for $\varepsilon \leq 0.5$. Remarkably, even when PATE-FM utilizes a stronger SemiSL pipeline, achieving higher non-private accuracy than our SelfSL pipeline, LabelDP-Pro still outperforms it at $\varepsilon = 0.18$ and $\varepsilon = 0.29$. We note that while LabelDP-Pro is a generic LabelDP algorithm, both the SelfSL-based approach and PATE-FM (SemiSL-based) heavily rely on powerful visual priors that can be derived from randomly augmented images alone. It is unclear how to generalize to other domains where there are no powerful input data augmentations. Therefore, while it is possible to plug LabelDP-Pro into a SemiSL pipeline as well by only privatizing (and denoising) the supervised loss gradients, we leave the evaluation of this complicated setup for future work.

## 6 EVALUATION WITH USER-LEVEL PRIVACY

We finally evaluate LabelDP-Pro with user-level privacy, which protects a user's entire contribution to the model instead of guaranteeing privacy for individual items. User-level DP offers more stringent but realistic protection against adversaries, especially in advertising applications.

**Dataset and model**    We use the Criteo Attribution Modeling for Bidding dataset[3] (Diemert Eustache, Meynet Julien et al., 2017), which comprises a 30-day sample of live traffic data from Criteo, a company that provides online display advertisements. Each example in the dataset corresponds to a banner impression shown to a user, and whether it resulted in a conversion attributed to Criteo. The dataset retains user identifiers, and each user may contribute to multiple examples. We randomly select $200,000$ users and gather their data to create the training dataset, and a disjoint set of $50,000$ users for evaluation. For the training data, we cap the maximum number of examples any one user can contribute to be $k$, where we vary $k$ in $\{2, 5, 10\}$[4]. In cases where a user contributes fewer than $k$ examples, we use random re-sampling to augment their contributions to $k$ examples. We train an attribution model (Appendix D.3) with cross-entropy loss and report the AUC. We use group privacy (see e.g. Vadhan, 2017) for randomized responses. For DP-SGD and LabelDP-Pro, we compute per-user gradients and use them to replace the per-example gradients used in traditional item-level DP-SGD. This inherently offers privacy guarantees at the user level (Kairouz et al., 2021a; Xu et al., 2023).

**Results**    We report the performance of LabelDP-Pro, RR, and DP-SGD[5]. As shown in Table 7, LabelDP-Pro consistently exhibits superior utility compared to RR when $\varepsilon < 5$, across various numbers of examples per user ($k$). As $k$ increases, the performance gap between RR and LabelDP-Pro becomes more pronounced. This widening gap can be attributed to the fact that a user-level DP guarantee of $(\varepsilon, \delta)$ translates into an item-level DP guarantee of $(\varepsilon/k, \delta/ke^\varepsilon)$ by group privacy (e.g., Vadhan (2017)). So as $k$ increases, the corresponding item-level privacy budget decreases, which further accentuates the advantage of LabelDP-Pro over RR as shown in Section 5.

## 7 CONCLUSIONS AND FUTURE WORK

We studied LabelDP-Pro, a family of label DP training algorithms that interleaves gradient projection steps with DP-SGD to improve the utility of the trained model. Our empirical study, encompassing item-level and user-level privacy, demonstrates that such projection-based algorithms improve the state-of-the-art utility in the high-privacy regime. We further support our empirical findings with theoretical analyses based on bias-variance trade-offs. An important direction is to

---

[3] https://ailab.criteo.com/criteo-attribution-modeling-bidding-dataset/

[4] The distribution of the number of examples contributed by each user is shown in Figure 5 (Appendix D.4).

[5] We do not include methods based on SemiSL or SelfSL in our analysis, because it is not clear how to effectively adapt those algorithms designed for image classification data to the domain of online advertising.

explore more efficient denoisers. Currently, the memory-efficient approach demands approximately 200 iterations to reach convergence, presenting a computation bottleneck within the overall training process. In addition, we aspire to investigate the potential of LabelDP-Pro in the context of regression tasks Ghazi et al. (2023), thereby expanding its applicability beyond the scope of the classification tasks that we study.

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

## A    RELATED WORK

Differential privacy (DP, Dwork et al., 2006b;a) is a widely adopted notion that could formally bound the privacy leakage of individual user information while still allowing for the analysis of population-level patterns. When it comes to training deep neural networks with DP guarantees, the go-to algorithm is DP-SGD, as introduced in the seminal work by Abadi et al. (2016). This algorithm operates by clipping the contributions of gradients on a per-example basis and injecting noise into the average gradient updates during each iteration of SGD.

Although several techniques have been proposed to enhance the utility of DP-SGD for specific downstream applications (Papernot et al., 2021; Li et al., 2022; Dörmann et al., 2021; Tramer & Boneh, 2021; Kurakin et al., 2022; Yu et al., 2022; De et al., 2022; Denison et al., 2023; Cummings et al., 2023), our work primarily focuses on utilizing the standard DP-SGD framework for the purpose of generalization. We anticipate that the utility improvements demonstrated in our work can be transferred to modern techniques.

The concept of label DP was initially introduced by Chaudhuri & Hsu (2011), who established lower bounds on the sample complexity of private PAC learners. More recently, there has been a growing interest in leveraging label-only privacy in the context of deep learning. The first work is by Ghazi et al. (2021), which introduced Label Privacy Multi-Stage Training (LP-MST), an approach for training deep neural classifiers while ensuring label DP. LP-MST operates by dividing the dataset into disjoint partitions across various stages of training. Notably, the soft output of the model from the previous stage serves as a prior, which is then used to improve the accuracy of the label randomizer in subsequent stages. In Appendix F.2, they presented a theoretical analysis of an algorithm referred to as LP-Normal-SGD, which involves projecting noisy gradients onto a subspace to improve noise scaling. This algorithm is captured by our LabelDP-Pro family. However, to the best of our knowledge, this algorithm has not been developed into a practical one.

Malek et al. (2021) introduced the idea that additive Laplace noise, combined with Bayesian inference, better aligns with the demands of learning tasks. Their contributions include the development of a soft RR algorithm called ALIBI and a more intricate framework named PATE-FM. PATE-FM combines the FixMatch semi-supervised learning algorithm (Sohn et al., 2020) with the PATE private learning framework (Papernot et al., 2018). Notably, their PATE-FM framework achieves state-of-the-art utility in image classification tasks. While the benefits of integrating unsupervised learning and semi-supervised techniques into the context of label DP have also been recognized by other studies, such as those conducted by Esfandiari et al. (2022); Tang et al. (2022), it is unclear if these approaches are immediately applicable to settings like online advertisement. This is primarily due to the ambiguity surrounding what constitutes an effective unsupervised feature extractor in such settings.

We also note that a few previous work has explored DP-SGD with gradient projection. Yu et al. (2021) proposed Gradient Embedding Perturbation (GEP), which firstly projects individual private gradient into a non-sensitive low-dimensional anchor subspace, and then perturbs the low-dimensional embedding and the residual gradient separately according to the privacy budget. However, Yu et al. (2021) requires a public dataset to compute their gradient subspace; while such data is possible for image classification tasks, it is typically not available for recommendation tasks, which is one of the major applications of LabelDP. Kairouz et al. (2021b) introduced a projection step to maintain the trajectory of the descent algorithm within the gradients' subspace. While this theoretically suggests faster convergence rates, empirical studies are not presented in the original work, making it challenging to establish a direct comparison.

## B    CENTRAL DP AND LOCAL DP

Recall the definition of $(\varepsilon, \delta)$-DP in Definition 1. This notion is usually referred to as the *central* DP when compared to an alternative notion of *local* DP defined below:

**Definition 5** (LocalDP; Raskhodnikova et al. (2008)). *Let $\varepsilon \in \mathbb{R}_{>0}, \delta \in [0, 1)$. A randomized algorithm $\mathcal{R}$ is said to be $(\varepsilon, \delta)$-LocalDP if for any two data points $z, z'$, and any subset $S$ of outputs of $\mathcal{R}$, it holds that $\Pr[\mathcal{R}(z) \in S] \leq e^{\varepsilon} \cdot \Pr[\mathcal{R}(z') \in S] + \delta$.*

In central DP, $\mathcal{A}$ acts as a trusted central server which has direct access to the dataset of sensitive examples and outputs (aggregated and) privatized results. On the other hand, LocalDP assumes that no central server can be trusted, and a randomized mechanism $\mathcal{R}$ runs locally by directly randomizing each individual data point. Since the outputs of $\mathcal{R}$ already provide DP guarantees, they can be sent to any (potentially malicious) central server for subsequent data analysis.

LocalDP can provide a privacy protection even when the trusted central server is compromised. However, the stringent guarantee usually requires a LocalDP algorithm to inject extremely high noise, severely compromising the utility of data analysis, especially in the high-privacy regime. On the other hand, in central DP, the randomized mechanism $\mathcal{A}$ takes in an entire dataset $D$, which allows it to compute a *distributional* property of $D$ without noise. By randomizing only the aggregated outcome, a central mechanism could better maintain the utility of distributional data analysis while still protecting the privacy of individual data points. In the context of machine learning, a typical example of LocalDP mechanism is the randomized response (RR) for LabelDP, and a typical example of central DP mechanism is DP-SGD.

## C  MEMORY-EFFICIENT PROJECTION VIA JACOBIAN VECTOR PRODUCT

We described a memory-efficient algorithm to compute the gradient projection in Section 3.2 using advanced autodiff primitives. While autodiff has been widely used in various domains such as atmospheric sciences and computational fluid dynamics, in most deep learning applications, only a special case of the reverse mode autodiff is used to compute the batch gradients. But as we discussed, the advanced autodiff primitives can be extremely useful to construct efficient gradient projection algorithms. We skip the details of how such primitives are realized and refer the curious readers to a survey by Baydin et al. (2018) and the references therein. Fortunately, modern deep learning frameworks such as PyTorch (Paszke et al., 2019), Tensorflow (Abadi et al., 2015), and JAX (Bradbury et al., 2018) are all starting to provide interfaces to such primitives to facilitate such advanced usage.

In the following we present a pseudocode implementation of the two subroutines $\mathbf{u} \mapsto \mathbf{G}\mathbf{u}$ for any $\mathbf{u} \in \mathbb{R}^{n_2 K}$, and $\mathbf{v} \mapsto \mathbf{G}^\top \mathbf{v}$ for any $\mathbf{v} \in \mathbb{R}^d$ that are required to compute $\mathbf{G}^\top(\mathbf{G}\boldsymbol{\alpha}_t - \tilde{\mathbf{g}})$. The pseudocode is written using the JAX functional style APIs, but can also be implemented using the corresponding APIs in PyTorch or Tensorflow.

```
def G_u(f, theta, x, u):
  """Computes G * u using VJP (grad).

  Args:
    f: the neural network function that maps inputs to logits
      given the model weights.
    theta: the neural network model weights.
    x: the (unlabeled) inputs.
    u: an array of shape (n2, K).

  Returns:
    The sum of per-example per-class gradients weighted by u,
    which is a nested array of the same structure as standard
    gradients.
  """

  def weighted_loss(theta):
    logits = f(theta, x)
    return - sum(u * log_softmax(logits, axis=-1))

  return jax.grad(weighted_loss)(theta)

def G_transpose_v(f, theta, x, v):
  """Compute G^T * v using JVP.
```

```
Args:
  f: the neural network function that maps inputs to logits
    given the model weights.
  theta: the model weights.
  x: the (unlabeled) inputs.
  v: a nested array of the same structure as the gradients.

Returns:
  An array of the shape (n2, K), where each entry is the inner
  product of each per-example per-class gradient with v.
"""

def raw_loss(theta):
  logits = f(theta, x)
  return -log_softmax(logits, axis=-1)

_, results = jax.jvp(raw_loss, (params,), (v,))
return results
```

## D  EXPERIMENTAL DETAILS AND MORE RESULTS

### D.1  BASELINE ALGORITHMS

**RR-Debiased**: RR-Debiased (originally proposed by Ghazi et al. (2021) and was referred to as LP-RR-SGD) is an extension of RR that uses unbiased loss for training. To elaborate, let us consider the multi-class classification setting where the input space is $\mathcal{X}$ and the label space $\mathcal{Y} = [K] :=$ $\{1, \ldots, K\}$. Given a training dataset $D \in \mathcal{X} \times \mathcal{Y}$ and a loss function $\ell : \mathbb{R}^K \times \mathcal{Y} \to \mathbb{R}$, our objective is to train a model $f : \mathcal{X} \to \mathbb{R}^K$ to learn the mapping from $\mathcal{X}$ to $\mathcal{Y}$.

When using RR for training, each example $(x, y) \in D$ undergoes a randomization process that produces a randomized label $\hat{y}$: Specifically, with a probability of $p := \frac{K}{e^\varepsilon + K - 1}$, $\hat{y}$ is set to a randomly drawn value from $[K]$, and with probability $1 - p$, $\hat{y}$ is set equal to $y$. However, the loss $\ell(f(x), \hat{y})$ introduces bias in estimating the true loss:

$$\mathbb{E}[\ell(f(x), \hat{y})] = (1 - p) \cdot \ell(f(x), y) + \frac{p}{K} \sum_{\kappa \in [K]} \ell(f(x), \kappa) \neq \ell(f(x), y).$$

To address this bias, RR-Debiased uses the following unbiased loss function $\hat{\ell}(f(x), \hat{y})$:

$$\hat{\ell}(f(x), \hat{y}) = \frac{\ell(f(x), \hat{y}) - \frac{p}{K} \sum_{\kappa \in [K]} \ell(f(x), \kappa)}{1 - p}.$$

We direct the readers to Appendix F.1 of Ghazi et al. (2021) for more details of this method.

**LP-2ST**: Label Privacy Multi-Stage Training (LP-MST, Ghazi et al., 2021) trains the model on disjoint partitions of the dataset in different stages. Specifically, our evaluation adopts LP-2ST, a specific configuration of LP-MST that employs two training stages, which is the primary setup evaluated in Ghazi et al. (2021). We use $40\%$ of the data for the first stage and $60\%$ for the second.

### D.2  EXPERIMENTAL DETAILS

**Hyper-parameters**  We search for the best hyper-parameter settings for each evaluated approach. For each set of hyper-parameters, we launch three independent runs with different random seeds. The final results are derived by averaging the accuracy results across these three runs, selecting the configuration that yields the highest performance. Table 8 provides the search space of hyper-parameters for results presented in the main paper. For LabelDP-Pro, we also search for hyper-parameters of the memory-efficient projection, including its learning rate in $\{0.01, 0.02, 0.05\}$, training step in $\{100, 200, 500\}$, and the label smoothing coefficient $\lambda$ in $\{0.7, 0.75, 0.8, 0.85\}$.

Table 8: Hyper-parameters for our experimental results presented in the main paper.

| | Optimizer | Learning rate | # Epochs | Batch size | Clipping norm (DP-SGD & LabelDP-Pro) |
|---|---|---|---|---|---|
| Table 5.a | SGD | $\{0.01, 0.05, 0.1, 0.2\}$ | $\{2, 5, 10\}$ | $\{1024\}$ | $\{1.0, 2.0, 5.0\}$ |
| Table 5.b | SGD | $\{0.01, 0.05, 0.1, 0.2\}$ | $\{50, 100\}$ | $\{1024\}$ | $\{1.0, 2.0, 5.0\}$ |
| Table 6 | SGD | $\{0.1, 0.2, 0.5, 1.0\}$ | $\{1, 2, 5, 10, 20\}$ | $\{1024\}$ | $\{1.0, 2.0, 5.0\}$ |
| Table 7 | RMSprop | $\{1e\text{-}4, 5e\text{-}4, 1e\text{-}3\}$ | $\{10, 20, 50\}$ | $\{256, 1024, 4096\}$ | $\{1.0, 2.0, 5.0\}$ |

**Self-supervised learning** For the self-supervised learning (SelfSL) experiments presented in Section 5.3, we pretrain the ResNet-18 (He et al., 2016) model on the CIFAR-10 dataset for $1,000$ epochs using SimCLR (Chen et al., 2020a). We use the Adam optimizer with a learning rate of $1 \times 10^{-3}$ (weight decay set to $1 \times 10^{-6}$), and a batch size of $1,024$. The pretraining takes around 26 hours on a single NVIDIA A100 GPU, with the pretraining loss trajectory presented in Figure 3.

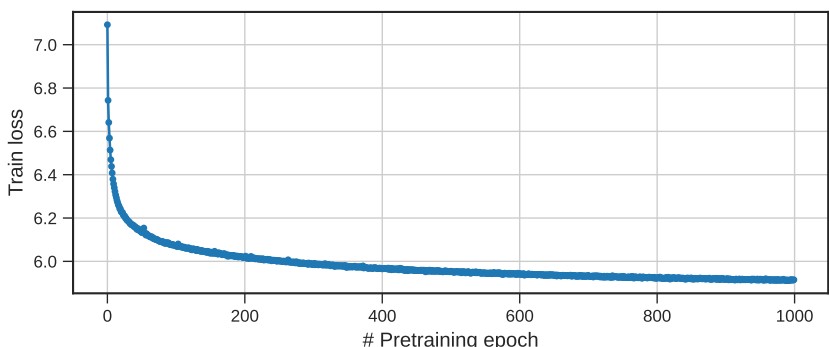

Figure 3: Pretraining loss of SelfSL with SimCLR (Chen et al., 2020a) on CIFAR-10.

For our comparison with RRWithPrior (Ghazi et al., 2021), we need to split the privacy budget for querying the prior and for training the model. The allocation of the budget for the prior across different $\varepsilon$ values is presented in Table 9.

Table 9: Budget for querying the prior in RRWithPrior for results in Table 6.

| $\varepsilon$ | Budget for prior querying |
|---|---|
| 0.01 | 0.005 |
| 0.02 | 0.01 |
| 0.05 | 0.02 |
| 0.18 | 0.05 |
| 0.29 | 0.05 |
| 0.5 | 0.1 |
| 1.6 | 0.1 |

### D.3 MODEL ARCHITECTURE

Table 10 provides the models used for item-level evaluation in Section 5.

For the experiments in user-level evaluation (Section 6), our feature set comprises a total of eleven features, including ten categorical features and one numerical feature. The categorical features are: `campaign`, a unique identifier for the campaign, and `cat[1-9]`, nine contextual features associated to the display. The numerical feature is `cost`, the price paid by Criteo for this display. Our goal is to predict the last-touch attribution, namely whether the final click on an impression leads to a conversion. Specifically, we firstly collect a subset of data where the impression has been clicked; we then create a new label named `last_touch_attribution_conversion`, which is defined as the Boolean AND of the original `conversion` label and (`click_pos == click_nb-1`). We then build a model to predict `last_touch_attribution_conversion` based on the features.

Table 10: Model architectures for item-level evaluation in Section 5.

(a) Model architecture for MNIST, FashionMNIST, and kMNIST

| Layer | Parameters |
|---|---|
| Conv1
AvgPool | in channels: 1, out channels: 16, kernel size: 3, activation: ReLU
kernel size: 2, stride: 2 |
| Conv2
AvgPool | in channels: 16, out channels: 16, kernel size: 3, activation: ReLU
kernel size: 2, stride: 2 |
| FC1
FC2 | out features: 16, activation: ReLU
out features: 10 |

(b) Model architecture for CIFAR-10

| Layer | Parameters |
|---|---|
| Conv1
MaxPool
Dropout | in channels: 3, out channels: 32, kernel size: 3, activation: ReLU
kernel size: 2, stride: 2
ratio: 0.2 |
| Conv2
MaxPool
Dropout | in channels: 32, out channels: 64, kernel size: 3, activation: ReLU
kernel size: 2, stride: 2
ratio: 0.2 |
| Conv3
MaxPool
Dropout | in channels: 64, out channels: 128, kernel size: 3, activation: ReLU
kernel size: 2, stride: 2
ratio: 0.2 |
| FC1
Dropout
FC2 | out features: 80, activation: ReLU
ratio: 0.2
out features: 10 |

## D.4 MORE RESULTS

**Running time** We provide the single-epoch training time for baseline methods and our LabelDP-Pro in Table 11, all implemented in JAX, utilizing a fixed batch size $n_1 = 1024$. For our method, we run 200 steps to solve the projection coefficients (see Section 3.2). Note that we have excluded the ALIBI method from our comparisons. This is because our experiments for ALIBI are based on the official ALIBI implementation in PyTorch[6], making direct runtime comparisons inappropriate. However, as presented in Table 7 of Tang et al. (2022), ALIBI should have a similar running time to LP-MST.

Table 11: Single-epoch training time (seconds) for different tasks. Results for MNIST, CIFAR-10, and CIFAR-10-SelfSL are reported on a single NVIDIA Tesla-P100 GPU, while results for Criteo Attribution are reported on a single NVIDIA A100 GPU.

| Task | RR | RR-debiased | LP-2ST | DP-SGD | LabelDP-Pro |
|---|---|---|---|---|---|
| MNIST (Table 5) | 2.40 | 88.34 | 331.30 | 44.52 | 59.93 |
| CIFAR-10 (Table 5) | 34.00 | 169.67 | 141.61 | 128.50 | 493.34 |
| CIFAR-10-SelfSL (Table 6) | 3.35 | 50.89 | - | 17.93 | 20.07 |
| Criteo Attribution (Table 7), $k = 5$ | 17.52 | - | - | 205.67 | 315.69 |

As shown, despite the iterative steps to solve the projection coefficient, LabelDP-Pro only incurs $1.11\times \sim 3.84\times$ computation overhead compared to DP-SGD. This is because we use `jax.lax.fori_loop()`, a JAX primitive that ensures the loops are compatible with just-in-time compilation, resulting in significant speedup gains. We also observe that the overhead is typically smaller for tasks with small models (e.g., MNIST-10 and CIFAR-10-SelfSL), potentially due to faster JVP because of lower weight dimension $d$.

**Intersection point** As demonstrated in Table 5, there appears to be a threshold $\varepsilon^*$ such that when $\varepsilon \leq \varepsilon^*$, LabelDP-Pro outperforms RR, but when $\varepsilon > \varepsilon^*$, RR exhibit superior performance. Table 12

---

[6]https://github.com/facebookresearch/label_dp_antipodes

demonstrates how the intersection point $\varepsilon^*$ varies with the number of classes ($K$) and the probability of catastrophic failure ($\delta$).

Our theoretical case study also demonstrates this phenomenon. When $e^\varepsilon \gg K$, we see that the additional excess error due to privacy Table 4 decays as $\sim \sqrt{K/e^\varepsilon}$. On the other hand, the same for LabelDP-Pro with ALTCONV denoiser decays as $\sim \log^{0.25}(K)/\sqrt{\varepsilon}$ (ignoring dependence on other parameters). Thus, for a fixed $K$, we expect RR-Debiased to outperform LabelDP-Pro for large enough $\varepsilon$.

- **Number of classes ($K$)**: We find that the larger the number of classes, the larger $\varepsilon^*$ becomes. This can be also be seen in our theoretical case study in Table 4, since the intersection point of $\sqrt{K/e^\varepsilon}$ and $\log^{0.25}(K)/\sqrt{\varepsilon}$ increases with increasing $K$.
- **Privacy parameter ($\delta$)**: Since LabelDP-Pro is an approximate-DP mechanism, we also explore the impact of varying $\delta$, which represents the probability of catastrophic failure, on $\varepsilon^*$. We find that when $\delta$ is smaller, $\varepsilon^*$ decreases. Indeed, the reduced tolerance for failure cases necessitates the injection of more noise into DP-SGD, ultimately leading to reduced utility in LabelDP-Pro.

Table 12: The intersection section point $\epsilon^*$ under different number of classes (i.e., $K$) and probability for catastrophic failure (i.e., $\delta$).

(a) $\varepsilon^*$ vs $K$

| $K$ | $\varepsilon^*$ |
|---|---|
| 2 | $1.0 \sim 1.1$ |
| 5 | $1.1 \sim 1.2$ |
| 10 | $1.2 \sim 1.3$ |

(b) $\varepsilon^*$ vs $\delta$

| $\delta$ | $\varepsilon^*$ |
|---|---|
| $1/n$ | $1.3 \sim 1.4$ |
| $1/n^{1.1}$ | $1.2 \sim 1.3$ |
| $1/n^{1.2}$ | $1.0 \sim 1.1$ |

**Item-level privacy** Table 13 presents results on CIFAR-10 with a larger range of $\varepsilon$. Table 14 and Table 15 present results on the kMNIST (Clanuwat et al., 2018) and FashionMNIST (Xiao et al., 2017) datasets. These results align with the findings presented in Table 5 of the main paper: we observe consistent improvements achieved by LabelDP-Pro over LabelDP in high-privacy settings, as well as over DP-SGD.

Table 13: Test accuracy of LabelDP, DP-SGD, and LabelDP-Pro on the CIFAR-10 dataset, for different $\varepsilon$'s. The best result for each $\varepsilon$ is **boldfaced**. LabelDP-Pro outperforms baseline LabelDP methods in the high-privacy (i.e., small $\varepsilon$) regime.

| Privacy Budget ($\varepsilon$) | Baseline LabelDP Methods | | | | DP-SGD | LabelDP-Pro |
|---|---|---|---|---|---|---|
| | RR | RR-DEBIASED | LP-2ST | ALIBI | | |
| 0.05 | 10.0 | 10.0 | 10.0 | 9.9 | 10.0 | **16.0** |
| 0.1 | 10.0 | 10.0 | 10.0 | 9.9 | 24.6 | **27.0** |
| 0.2 | 10.0 | 10.0 | 10.0 | 17.9 | 29.8 | **30.8** |
| 0.5 | 10.0 | 10.0 | 35.8 | 34.4 | 36.9 | **40.2** |
| 1.0 | 17.0 | 10.0 | **60.9** | 51.3 | 43.5 | 44.8 |
| 2.0 | 77.6 | 58.8 | 63.8 | **64.2** | 51.4 | 52.0 |
| 5.0 | 80.4 | 80.2 | 81.7 | **82.3** | 59.5 | 59.8 |
| 10.0 | 81.5 | 81.8 | 83.5 | **83.5** | 63.6 | 63.7 |

Table 14: Test accuracy of LabelDP, DP-SGD, and LabelDP-Pro on the FashionMNIST dataset, for different $\varepsilon$'s. The best result for each $\varepsilon$ is **boldfaced**. LabelDP-Pro outperforms baseline LabelDP methods in the high-privacy (i.e., small $\varepsilon$) regime.

| Privacy Budget ($\varepsilon$) | Baseline LabelDP Methods | | | | DP-SGD | LabelDP-Pro |
|---|---|---|---|---|---|---|
| | RR | RR-DEBIASED | LP-2ST | ALIBI | | |
| 0.01 | 13.9 | 10.0 | 10.0 | 9.9 | 59.1 | **63.0** |
| 0.1 | 15.7 | 10.0 | 10.0 | 10.7 | 73.4 | **75.7** |
| 0.2 | 19.8 | 10.1 | 10.0 | 10.7 | 75.6 | **77.9** |
| 0.5 | 46.9 | 44.8 | 10.9 | 69.2 | 79.7 | **79.8** |
| 1.0 | 75.7 | 73.9 | 68.7 | 76.5 | 81.5 | **81.7** |
| 2.0 | 84.6 | **85.0** | 84.8 | 82.3 | 82.3 | 82.4 |

Table 15: Test accuracy of LabelDP, DP-SGD, and LabelDP-Pro on the kMNIST dataset, for different $\varepsilon$'s. The best result for each $\varepsilon$ is **boldfaced**. LabelDP-Pro outperforms baseline LabelDP methods in the high-privacy (i.e., small $\varepsilon$) regime.

| Privacy Budget ($\varepsilon$) | Baseline LabelDP Methods | | | | DP-SGD | LabelDP-Pro |
|---|---|---|---|---|---|---|
| | RR | RR-DEBIASED | LP-2ST | ALIBI | | |
| 0.01 | 7.6 | 10.0 | 10.0 | 10.0 | 40.6 | **43.4** |
| 0.1 | 8.2 | 10.0 | 10.0 | 10.8 | 61.1 | **62.1** |
| 0.2 | 10.0 | 10.0 | 10.0 | 10.9 | 64.8 | **67.4** |
| 0.5 | 10.0 | 10.1 | 12.2 | 49.7 | 68.7 | **72.4** |
| 1.0 | 31.1 | 28.9 | 55.1 | 69.8 | 72.8 | **74.5** |
| 2.0 | 82.4 | 81.8 | **83.2** | 82.8 | 75 | 76.3 |

**RR results in SelfSL**  We notice that when $\varepsilon < 1.6$, combining RR with SelfSL results in a surprisingly low accuracy (see Table 6), even worse than the performance of random guessing. To investigate the reason, we visualize the confusion matrices on test data for the initialized model, and for trained models with RR with $\varepsilon = 0.1$ and $\varepsilon = 1.6$ in Figure 4. We observed that at initialization, the predictions are biased towards a few labels such as '1', '5', '6', '7', suggesting that the representations from SelfSL are probably not very balanced. With the randomly initialized weights, on average the prediction accuracy is around 10% as expected. However, we observe that after training, for $\varepsilon = 0.1$, the model is severely confused by the strong noises in the training labels. It maintains a strong bias to predict '1', '6' and '7', but only when the true labels are not '1', '6', or '7', respectively. As a result, the accuracy is even lower than random guessing (10%). In contrast, when $\varepsilon$ is set to a larger value (say, $\varepsilon = 1.6$) which returns the true label with higher probability, the model's prediction aligns better with the ground truth label.

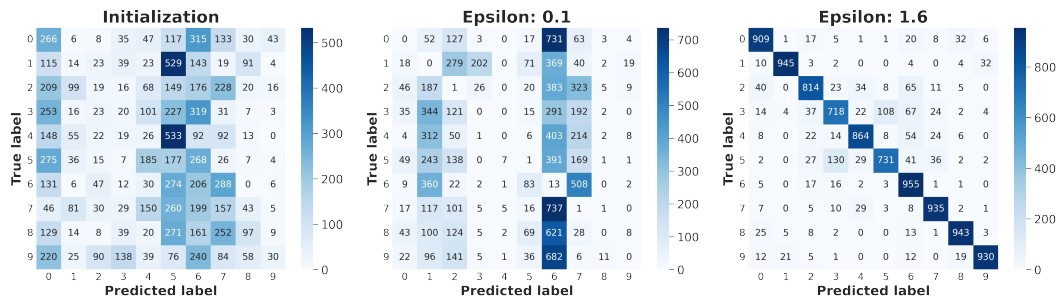

Figure 4: Confusion matrices computed on test data for the randomly initialized model (left), and for trained models with RR using $\varepsilon = 0.1$ (middle) and $\varepsilon = 1.6$ (right). When $\varepsilon$ is small, the model is highly biased towards predicting an unseen example as '1', '6', and '7'.

**User-level privacy**  Figure 5 presents the distribution of the number of examples contributed per user in the Criteo Attribution dataset.

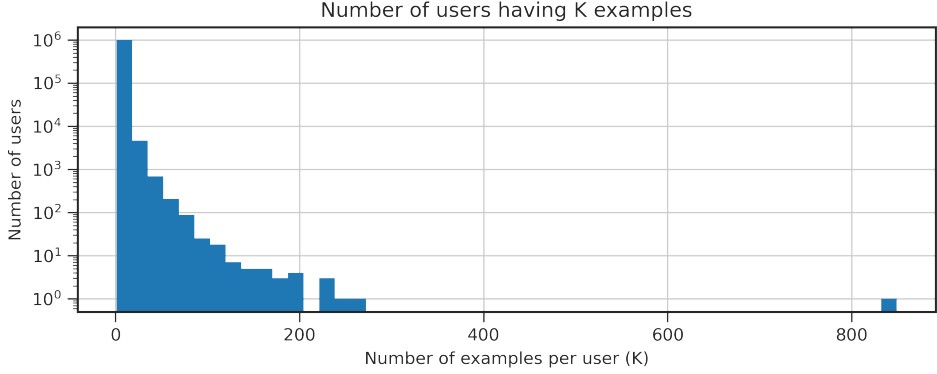

Figure 5: Examples per user in the Criteo Attribution dataset.

# E PROOFS FOR THEORETICAL CASE STUDY (SECTION 4)

## E.1 EXCESS ERROR IN STOCHASTIC CONVEX OPTIMIZATION

For completeness, we prove Lemma 3, restated below for convenience. The proof follows from standard analysis of projected gradient descent (see, e.g., Hazan (2022)).

**Lemma 3.** *For a convex loss function $\mathcal{L}$, and a gradient oracle with bias $\alpha$ and expected squared norm $\widetilde{C}^2$, PGD over a convex set $\mathcal{K} \subseteq \mathbb{R}^d$ with diameter $R$ and step size $\eta = \frac{R}{\widetilde{C}\sqrt{T}}$ achieves the following expected excess risk bound for the average iterate $\bar{\boldsymbol{w}}_T := \frac{1}{T}\sum_{t=0}^{T-1}\boldsymbol{w}_t$:*

$$\mathbb{E}\left[\mathcal{L}\left(\bar{\boldsymbol{w}}_T\right)\right] - \mathcal{L}(\boldsymbol{w}^*) \;\leq\; \frac{R\widetilde{C}}{\sqrt{T}} + \alpha R. \tag{1}$$

*Proof.* Suppose we have a gradient oracle with bias $\alpha$ and expected squared norm $\widetilde{C}^2$. Let $\boldsymbol{g}_t = \nabla\mathcal{L}(\boldsymbol{w}_t)$ denote the true gradient, and let $\tilde{\boldsymbol{g}}_t$ denote the stochastic gradient received at step $t$. Let $\boldsymbol{e}_t = \boldsymbol{g}_t - \tilde{\boldsymbol{g}}_t$ be the error in the stochastic gradient. We have that $\|\mathbb{E}\,\boldsymbol{e}_t\|_2 \leq \alpha$ and $\mathbb{E}\,\|\tilde{\boldsymbol{g}}_t\|_2^2 \leq \widetilde{C}^2$. Recall that PGD operates by computing $\boldsymbol{w}_{t+1} \leftarrow \mathsf{Proj}_{\mathcal{K}}(\boldsymbol{w}_t - \eta\tilde{\boldsymbol{g}}_t)$. We have

$$
\begin{aligned}
&\mathcal{L}(\boldsymbol{w}_t) - \mathcal{L}(\boldsymbol{w}^*) \\
&\leq \langle \boldsymbol{g}_t, \boldsymbol{w}_t - \boldsymbol{w}^* \rangle \\
&= \langle \boldsymbol{e}_t, \boldsymbol{w}_t - \boldsymbol{w}^* \rangle + \langle \tilde{\boldsymbol{g}}_t, \boldsymbol{w}_t - \boldsymbol{w}^* \rangle \\
&= \langle \boldsymbol{e}_t, \boldsymbol{w}_t - \boldsymbol{w}^* \rangle + \tfrac{1}{\eta} \cdot \langle \eta\tilde{\boldsymbol{g}}_t, \boldsymbol{w}_t - \boldsymbol{w}^* \rangle \\
&= \langle \boldsymbol{e}_t, \boldsymbol{w}_t - \boldsymbol{w}^* \rangle + \tfrac{1}{2\eta} \cdot \left( \|\boldsymbol{w}_t - \boldsymbol{w}^*\|_2^2 + \eta^2\|\tilde{\boldsymbol{g}}_t\|_2^2 - \|\boldsymbol{w}_t - \eta\tilde{\boldsymbol{g}}_t - \boldsymbol{w}^*\|_2^2 \right) \\
&\leq \langle \boldsymbol{e}_t, \boldsymbol{w}_t - \boldsymbol{w}^* \rangle + \tfrac{1}{2\eta} \cdot \left( \|\boldsymbol{w}_t - \boldsymbol{w}^*\|_2^2 + \eta^2\|\tilde{\boldsymbol{g}}_t\|_2^2 - \|\boldsymbol{w}_{t+1} - \boldsymbol{w}^*\|_2^2 \right) \\
&= \langle \boldsymbol{e}_t, \boldsymbol{w}_t - \boldsymbol{w}^* \rangle + \tfrac{\eta\|\tilde{\boldsymbol{g}}_t\|_2^2}{2} + \tfrac{1}{2\eta} \cdot \left( \|\boldsymbol{w}_t - \boldsymbol{w}^*\|_2^2 - \|\boldsymbol{w}_{t+1} - \boldsymbol{w}^*\|_2^2 \right),
\end{aligned}
$$

where the second inequality is due to the fact that $\boldsymbol{w}_{t+1}$ is a projection of $\boldsymbol{w}_t - \eta\tilde{\boldsymbol{g}}_t$ onto $\mathcal{K}$ and $\boldsymbol{w}^* \in \mathcal{K}$, and hence $\|\boldsymbol{w}_{t+1} - \boldsymbol{w}^*\| \leq \|\boldsymbol{w}_t - \eta\tilde{\boldsymbol{g}}_t - \boldsymbol{w}^*\|$.

Averaging over $t = 0, \ldots, T-1$ and using Jensen's inequality we have for $\bar{\boldsymbol{w}}_T := \frac{1}{T}\sum_{t=0}^{T-1}\boldsymbol{w}_t$ that

$$
\begin{aligned}
\mathcal{L}(\bar{\boldsymbol{w}}_T) - \mathcal{L}(\boldsymbol{w}^*) &\leq \tfrac{1}{T}\sum_{t=0}^{T-1}\mathcal{L}(\boldsymbol{w}_t) - \mathcal{L}(\boldsymbol{w}^*) \\
&\leq \tfrac{1}{2\eta T} \cdot \|\boldsymbol{w}_0 - \boldsymbol{w}^*\|_2^2 + \tfrac{\eta}{2T}\sum_{t=0}^{T-1}\|\tilde{\boldsymbol{g}}_t\|_2^2 + \tfrac{1}{T}\sum_{t=0}^{T-1}\langle \boldsymbol{e}_t, \boldsymbol{w}_t - \boldsymbol{w}^* \rangle \\
&\leq \tfrac{R^2}{2\eta T} + \tfrac{\eta}{2T}\sum_{t=0}^{T-1}\|\tilde{\boldsymbol{g}}_t\|_2^2 + \tfrac{1}{T}\sum_{t=0}^{T-1}\langle \boldsymbol{e}_t, \boldsymbol{w}_t - \boldsymbol{w}^* \rangle.
\end{aligned}
$$

Taking expectation on both sides and using the convexity of $\mathcal{L}$, we have

$$
\begin{aligned}
\mathbb{E}[\mathcal{L}(\bar{\boldsymbol{w}}_T) - \mathcal{L}(\boldsymbol{w}^*)] &\leq \frac{R^2}{2\eta T} + \frac{\eta\widetilde{C}^2}{2} + \frac{1}{T}\sum_{t=0}^{T-1}\mathbb{E}\left[\langle \boldsymbol{e}_t, \boldsymbol{w}_t - \boldsymbol{w}^* \rangle\right] \\
&\leq \frac{R^2}{2\eta T} + \frac{\eta\widetilde{C}^2}{2} + \frac{1}{T}\sum_{t=0}^{T-1}R \cdot \|\mathbb{E}\,\boldsymbol{e}_t\|_2 \\
&\leq \frac{R^2}{2\eta T} + \frac{\eta\widetilde{C}^2}{2} + \alpha R \\
&\leq \frac{R\widetilde{C}}{\sqrt{T}} + \alpha R \qquad \text{for } \eta = \frac{R}{\widetilde{C}\sqrt{T}}. \qquad \square
\end{aligned}
$$

## E.2 ERROR BOUNDS OF STOCHASTIC GRADIENT ORACLES

We analyze the error introduced by various stochastic gradient oracles thereby using Lemma 3 to obtain excess rates for projected gradient descent. First, we observe that just sampling a mini-batch $I^G$ already introduces some variance in the stochastic gradients, which can be bounded as follows for $C$-Lipschitz losses.

We use the following notation in this section. Let $\boldsymbol{g} = \mathcal{L}(\boldsymbol{w}) := \frac{1}{n} \sum_{i \in [n]} \nabla_{\boldsymbol{w}} \ell(\boldsymbol{w}; x_i, y_i)$. For a randomly sampled mini-batch $I^G \subseteq [n]$ with $|I^G| = n_1$, let $\hat{\boldsymbol{g}}^1 := \frac{1}{n_1} \sum_{i \in I^G} \nabla_{\boldsymbol{w}} \ell(\boldsymbol{w}; x_i, y_i)$.

**Lemma 6.** *Over the choice of a random subset $I^G \subseteq [n]$ with $|I^G| = n_1$, it holds that*

$$\mathbb{E}\, \hat{\boldsymbol{g}}^1 = \boldsymbol{g} \qquad and \qquad \mathbb{E}\left[\|\hat{\boldsymbol{g}}^1 - \boldsymbol{g}\|_2^2\right] \leq \frac{C^2}{n_1}.$$

*Proof.* The first part holds by the linearity of expectation. To show the second part, let $\boldsymbol{v}_i := \nabla_{\boldsymbol{w}} \ell(\boldsymbol{w}; x_i, y_i) - \boldsymbol{g}$. We have that $\sum_{i \in [n]} \boldsymbol{v}_i = 0$. Hence for two indices $i, j$ sampled without replacement from $[n]$, it holds that $\mathbb{E}\left[\langle \boldsymbol{v}_i, \boldsymbol{v}_j \rangle\right] < 0$, since $\mathbb{E}[\boldsymbol{v}_i | \boldsymbol{v}_j] = -\boldsymbol{v}_j/(n-1)$. Hence we have

$$
\begin{aligned}
\mathbb{E}\left[\|\hat{\boldsymbol{g}}^1 - \boldsymbol{g}\|_2^2\right] &= \mathbb{E}\left[\left\|\frac{1}{n_1} \sum_{i \in I^G} \boldsymbol{v}_i\right\|^2\right] \\
&= \frac{1}{n_1^2}\left(\sum_{i \in I^G} \mathbb{E}[\|\boldsymbol{v}_i\|^2] + 2\sum_{i \neq j \in I^G} \underbrace{\mathbb{E}[\langle \boldsymbol{v}_i, \boldsymbol{v}_j \rangle]}_{\leq 0}\right) \leq \frac{C^2}{n_1}. \qquad \square
\end{aligned}
$$

We now analyze the excess error introduced by stochastic gradient oracles considered in Table 4, and explicitly focus on the *additional excess error* that is introduced due to privacy. In order to do so, we note that the non-private mini-batch gradient averages have a variance of $\frac{C^2}{n_1}$, due to which the PGD with this oracle achieves an excess error bound of

$$O\left(\frac{RC}{\sqrt{T}}\left(1 + \frac{1}{\sqrt{n_1}}\right)\right). \tag{2}$$

The instantiations of LabelDP-Pro operate by obtaining a noisy gradient estimate $\tilde{\boldsymbol{g}} = \hat{\boldsymbol{g}}^1 + \boldsymbol{e}$ for $\boldsymbol{e} \sim \mathcal{N}(0, \frac{\sigma^2 C^2}{n_1} \cdot \mathbf{I})$, and then potentially projecting it. The addition excess error incurred due to privacy can be deduced by subtracting (2) from the error bound.

**NoOp (same as DP-SGD).** The vanilla DP-SGD algorithm performs PGD using a stochastic gradient with zero bias (i.e., $\alpha = 0$) and variance $\omega^2 = \frac{C^2}{n_1} + \frac{d\sigma^2 C^2}{n_1}$, where recall that $d$ is the dimension of the parameter $\boldsymbol{w}$. The first term of the variance comes from Lemma 6 and the second term is due to the addition of noise. Using Lemma 3 and subtracting (2), this implies that the additional excess error bound due to privacy is

$$O\left(\frac{RC}{\sqrt{T}} \cdot \frac{\sqrt{d}\sigma}{\sqrt{n_1}}\right).$$

**SelfSpan.** Recall that this stochastic gradient oracle returns $\tilde{\boldsymbol{z}} = \mathsf{Proj}_A(\tilde{\boldsymbol{g}})$ where $A = \mathsf{span}\left(\{\nabla_{\boldsymbol{w}} \ell(\boldsymbol{w}, (x_i, \kappa)\}_{i \in I^G, \kappa \in [K]}\right)$. Since $\hat{\boldsymbol{g}}^1 \in A$, we have that $\tilde{\boldsymbol{z}} = \hat{\boldsymbol{g}}^1 + \mathsf{Proj}_A(\boldsymbol{e})$. This span has dimension at most $Kn_1$, which can be much smaller than $d$. Hence, this stochastic gradient oracle has zero bias and variance $\omega^2 = \frac{C^2}{n_1} + \frac{Kn_1\sigma^2 C^2}{n_1}$. Using Lemma 3 and subtracting (2), this implies an additional excess error bound due to privacy of

$$O\left(\frac{RC}{\sqrt{T}} \cdot \sqrt{K}\sigma\right).$$

Thus, we have reduced the $\sqrt{d/n_1}$ term in excess error rate to $\sqrt{K}$. However, this comes at a price! Unlike DP-SGD, we cannot leverage amplification by subsampling because the (unlabeled) examples were used to define the span on which the noisy gradient was projected. Thus, the noise multiplier $\sigma$ required for the same privacy guarantee is (roughly $\sqrt{n/n_1}$ times) larger, due to which this bound is incomparable to the bound of DP-SGD.

**SelfConv.** Recall that this stochastic gradient oracle returns $\tilde{\boldsymbol{z}} = \mathsf{Proj}_A(\tilde{\boldsymbol{g}})$ where $A = \mathsf{conv}\left(\{\nabla_{\boldsymbol{w}} \ell(\boldsymbol{w}, (x_i, \kappa)\}_{i \in I^G, \kappa \in [K]}\right)$. We show the following.

**Lemma 7.** *Let $I^G \subseteq [n]$ be a random subset of size $n_1$. Let $\tilde{\boldsymbol{g}}$ denote the noisy gradient estimate from $I^G$, namely, $\tilde{\boldsymbol{g}} := \frac{1}{n_1} \sum_{i \in I^G} \nabla_{\boldsymbol{w}} \ell(\boldsymbol{w}; x_i, y_i) + \boldsymbol{e}$ for $\boldsymbol{e} \sim \mathcal{N}(0, \frac{\sigma^2 C^2}{n_1} \cdot \mathbf{I})$. Let $A = \mathsf{conv}\left(\{\nabla_{\boldsymbol{w}} \ell(\boldsymbol{w}, (x_i, \kappa)\}_{i \in I^G, \kappa \in [K]}\right)$ denote the convex body on which we project. Then, over*

*the choice of $I^G$ and $e$, we have*

$$\mathbb{E}\left[\|\operatorname{Proj}_A(\tilde{g}) - g\|_2^2\right] \leq O\left(C^2\left(\frac{1}{n_1} + \frac{\sigma\sqrt{\log(Kn_1)}}{\sqrt{n_1}}\right)\right).$$

*Proof.* Let $\tilde{z} := \operatorname{Proj}_A(\tilde{g})$. Since $A$ is a convex set, we have $\langle \tilde{z} - v, \tilde{z} - \tilde{g}\rangle \leq 0$, for all $v \in A$. By definition, we have that $\hat{g}^1 \in A$ and hence setting $v = \hat{g}^1$ and rearranging, we arrive at

$$\langle \tilde{z} - \hat{g}^1, \tilde{z} - \hat{g}^1\rangle \leq \langle \tilde{z}, e\rangle$$
$$\leq \max\{|\langle v, e\rangle| : v \in A\} \qquad \text{(since } \tilde{z} \in A\text{)}$$
$$= \max\{|\langle \nabla_w \ell(w; x_i, \kappa), e\rangle| : i \in B_2, \kappa \in [K]\}.$$

Thus, $\qquad \mathbb{E}\left[\|\tilde{z} - \hat{g}^1\|_2^2\right] \leq O\left(\frac{\sigma C^2\sqrt{\log(Kn_1)}}{\sqrt{n_1}}\right),$

which follows by observing that $\|\nabla_w \ell(w; x_i, \kappa)\| \leq C$ and using the standard bound on the Gaussian mean width of a convex hull of a finite set. Thus, by the triangle inequality and using Lemma 6, we have that

$$\mathbb{E}\left[\|\tilde{z} - g\|_2^2\right] \leq 2\mathbb{E}\left[\|\hat{g}^1 - g\|_2^2 + \|\tilde{z} - \hat{g}^1\|_2^2\right] \leq O\left(\frac{C^2}{n_1} + \frac{\sigma C^2\sqrt{\log(Kn_1)}}{\sqrt{n_1}}\right). \qquad \square$$

Thus it follows that the SELFCONV stochastic gradient oracle has bias $\alpha$ and error at most $O\left(C \cdot \left(\frac{1}{\sqrt{n_1}} + \frac{\sqrt{\sigma}\sqrt[4]{\log(Kn_1)}}{\sqrt[4]{n_1}}\right)\right)$. Moreover the expected square gradient norm $\widetilde{C} \leq C$ (since all vectors in the convex hull have norm at most $C$). Using Lemma 3 and subtracting (2), this implies an additional excess error bound due to privacy of

$$O\left(RC\left(\frac{1}{\sqrt{n_1}} + \frac{\sqrt{\sigma}\sqrt[4]{\log(Kn_1)}}{\sqrt[4]{n_1}}\right)\right).$$

But as in SELFSPAN, we cannot leverage amplification by subsampling and the noise multiplier $\sigma$ here is (again, roughly $\sqrt{n/n_1}$ times) larger as compared to DP-SGD. However, now we have a non-zero bias, and hence a non-vanishing (in $T$) term.

**ALTCONV.** Recall that this stochastic gradient oracle returns $\tilde{z} = \operatorname{Proj}_A(\tilde{g})$ where $A = \operatorname{conv}\left(\{\nabla_w \ell(w, (x_i, \kappa)\}_{i \in I^P, \kappa \in [K]}\right)$. We show Lemma 4 restated below for convenience.

**Lemma 4.** *Let $I^G, I^P \subseteq [n]$ be random subsets of size $n_1, n_2$ respectively. Let $\tilde{g}$ denote the noisy gradient estimate from $I^G$, namely, $\tilde{g} := \frac{1}{n_1}\sum_{i \in I^G}\nabla_w \ell(w; x_i, y_i) + e$ for $e \sim \mathcal{N}(0, \frac{\sigma^2 C^2}{n_1}\cdot \mathbf{I})$. Let $A = \operatorname{conv}\left(\{\nabla_w \ell(w, (x_i, \kappa))\}_{i \in I^P, \kappa \in [K]}\right)$ denote the convex body on which we project. Then, over the choice of $I^G, I^P$, and $e$, we have*

$$\mathbb{E}\left[\|\operatorname{Proj}_A(\tilde{g}) - g\|_2^2\right] \leq O\left(C^2 \cdot \left(\frac{1}{n_1} + \frac{1}{n_2} + \frac{\sigma\sqrt{\log(Kn_2)}}{\sqrt{n_1}}\right)\right).$$

*Proof.* Let $\hat{g}^1, \hat{g}^2$ denote the gradient estimates from $I^G, I^P$ respectively. Formally,
$$\hat{g}^1 := \frac{1}{n_1}\sum_{i \in I^G}\nabla_w \ell(w; x_i, y_i), \qquad \hat{g}^2 := \frac{1}{n_2}\sum_{i \in I^P}\nabla_w \ell(w; x_i, y_i).$$
Let $\tilde{z} := \operatorname{Proj}_A(\tilde{g})$. Since $A$ is a convex set, we have:
$$\langle \tilde{z} - v, \tilde{z} - \tilde{g}\rangle \leq 0, \qquad \text{for all } v \in A.$$
By definition, we have that $\hat{g}^2 \in A$ and hence setting $w = \hat{g}^2$ and rearranging, we arrive at
$$\langle \tilde{z} - \hat{g}^2, \tilde{z} - \hat{g}^1\rangle \leq \langle \tilde{z}, e\rangle$$
$$\leq \max\{|\langle v, e\rangle| : v \in A\} \qquad \text{(since } \tilde{z} \in A\text{)}$$
$$= \max\{|\langle \nabla_w \ell(w; x_i, y), e\rangle| : i \in B_2, y \in [K]\}.$$

Thus, $\qquad \mathbb{E}\left[\langle \tilde{z} - \hat{g}^2, \tilde{z} - \hat{g}^1\rangle\right] \leq O\left(\frac{\sigma C^2\sqrt{\log(Kn_2)}}{\sqrt{n_1}}\right),$

which follows by observing that $\|\nabla_{\boldsymbol{w}}\ell(\boldsymbol{w};x_i,\kappa)\| \leq C$ and using the standard bound on Gaussian mean width of a convex hull of a finite set. Thus, we have that

$$
\begin{aligned}
\mathbb{E}\left[\|\tilde{\boldsymbol{z}} - \hat{\boldsymbol{g}}^1\|_2^2\right] &\leq \mathbb{E}\left[\|\tilde{\boldsymbol{z}} - \hat{\boldsymbol{g}}^1\|^2 + \|\tilde{\boldsymbol{z}} - \hat{\boldsymbol{g}}^2\|^2\right] \\
&= \mathbb{E}\left[\|\hat{\boldsymbol{g}}^1 - \hat{\boldsymbol{g}}^2\|^2 + 2\left\langle \tilde{\boldsymbol{z}} - \hat{\boldsymbol{g}}^1, \tilde{\boldsymbol{z}} - \hat{\boldsymbol{g}}^2 \right\rangle\right] \\
&\leq O\left(\frac{C^2}{n_1} + \frac{C^2}{n_2} + \frac{\sigma C^2\sqrt{\log(Kn_2)}}{\sqrt{n_1}}\right),
\end{aligned}
$$

where the last step uses Lemma 6. Thus, we have by the triangle inequality that

$$
\mathbb{E}\left[\|\tilde{\boldsymbol{z}} - \boldsymbol{g}\|_2^2\right] \leq 2\,\mathbb{E}\left[\|\tilde{\boldsymbol{z}} - \hat{\boldsymbol{g}}^1\|_2^2 + \|\hat{\boldsymbol{g}}^1 - \boldsymbol{g}\|_2^2\right] \leq O\left(C^2 \cdot \left(\frac{1}{n_1} + \frac{1}{n_2} + \frac{\sigma\sqrt{\log(Kn_2)}}{\sqrt{n_1}}\right)\right). \quad \square
$$

Moreover the expected squared gradient norm $\widetilde{C}^2 \leq C^2$ (since all vectors in the convex hull have norm at most $C$). Using Lemma 3 and subtracting (2), this implies an additional excess error bound due to privacy of

$$
O\left(RC \cdot \left(\frac{1}{\sqrt{n_1}} + \frac{1}{\sqrt{n_2}} + \frac{\sqrt{\sigma}\sqrt[4]{\log(Kn_2)}}{\sqrt[4]{n_1}}\right)\right).
$$

While we again have a non-zero bias, and hence a non-vanishing term (in $T$), we can leverage amplification by subsampling!

**RR-Debiased.** Recall from Appendix D.1, that RR-Debiased uses a gradient oracle that first samples $i \in [n]$, and using $p = \frac{K}{e^\varepsilon + K - 1}$, samples a random label $\hat{y}_i$ as

$$
\Pr[\hat{y}_i = y] = \begin{cases} 1 - \frac{p(K-1)}{K} & \text{if } \hat{y}_i = y_i \\ \frac{p}{K} & \text{if } \hat{y}_i \neq y_i, \end{cases}
$$

and then returns the stochastic gradient

$$
\tilde{\boldsymbol{g}} = \frac{\nabla_{\boldsymbol{w}}\ell(\boldsymbol{w};x_i,\hat{y}_i) - \frac{p}{K}\sum_{\kappa \in [K]}\nabla_{\boldsymbol{w}}\ell(\boldsymbol{w};x_i,\kappa)}{1-p}.
$$

For ease of notation, let $\boldsymbol{g}_{x,y} := \nabla_{\boldsymbol{w}}\ell(\boldsymbol{w};x,y)$. It is easy to see (also shown in Ghazi et al. (2021)), that this stochastic gradient has zero bias (i.e., $\mathbb{E}[\tilde{\boldsymbol{g}}] = \boldsymbol{g}$). Moreover, the variance of the stochastic gradient introduced by the randomized response for any fixed $(x_i, y_i)$ is given as

$$
\begin{aligned}
&\mathbb{E}\left[\|\tilde{\boldsymbol{g}} - \boldsymbol{g}_{x_i,y_i}\|_2^2\right] \\
&= \frac{1}{(1-p)^2}\,\mathbb{E}\left[\left\|\boldsymbol{g}_{x_i,\hat{y}_i} - \frac{p}{K}\sum_{\kappa \in [K]}\boldsymbol{g}_{x_i,\kappa} - (1-p)\boldsymbol{g}_{x_i,y_i}\right\|_2^2\right] \\
&= \frac{1}{(1-p)^2} \cdot \mathbb{E}\left[\begin{array}{l}(1 - p + \frac{p}{K}) \cdot \|p\cdot(1-\frac{1}{K})\boldsymbol{g}_{x_i,y_i} - \frac{p}{K}\sum_{\kappa \neq y_i}\boldsymbol{g}_{x_i,\kappa}\|_2^2 \\ + \frac{p}{K}\sum_{\kappa' \neq i}\|(1-\frac{p}{K})\boldsymbol{g}_{x_i,\kappa'} - \frac{p}{K}\sum_{\kappa \neq \kappa',i}\boldsymbol{g}_{x_i,\kappa} - (1 - p + \frac{p}{K})\boldsymbol{g}_{x_i,y_i}\|_2^2\end{array}\right] \\
&\leq O\left(\frac{p}{(1-p)^2} \cdot C^2\right).
\end{aligned}
$$

Note that Ghazi et al. (2021) also establish a bound on the variance, however, they state an approximate version that is only applicable when $\varepsilon \ll 1$; whereas, the above bound holds for all $\varepsilon$. In fact, their version follows from the bound above, since for $\varepsilon \ll 1$, $p \approx 1$ and $1 - p \approx \varepsilon/K$, and thus the bound becomes $O(C^2K^2/\varepsilon^2)$. On the other hand, when $e^\varepsilon \gg K$, we can approximate $p \approx K/e^\varepsilon$ and $1 - p \approx 1$, and hence the bound is $O(C^2K/e^\varepsilon)$.

Finally, when using a mini-batch $I^G$ with $|I^G| = n_1$, the variance due to randomized response reduces by a factor of $1/n_1$, and hence using Lemma 3 and subtracting (2), this implies an additional excess error bound due to privacy of

$$
O\left(\frac{RC}{\sqrt{T}} \cdot \frac{\sqrt{p}}{(1-p)\sqrt{n_1}}\right).
$$

