# OpenReview forum: "LabelDP-Pro: Learning with Label Differential Privacy via Projections"
_ICLR.cc/2024/Conference — ICLR 2024 poster_

### Official Review · Reviewer_NHnW · 2023-10-30

**Soundness:** 3 good
**Presentation:** 3 good
**Contribution:** 3 good
**Rating:** 6
**Confidence:** 4

**Summary:**

This paper introduces a cute idea -- leveraging the feature vectors (which are non-private under label DP) to construct a prior, which helps to reduce the noise required for label DP. To this end, this paper considers label-DP in the central model rather than the previous local model. Some theoretical results and experimental results are provided.

**Strengths:**

1. The idea of leveraging feature vectors to boost the performance under the label DP is cute.
2. A good balance between theory and experimental results, which is a good fit for ICLR.

**Weaknesses:**

I think the comparisons in the theoretical part need to be clear, i.e., what's the exact gain of the proposed method?

**Questions:**

I think this paper introduces some nice ideas and I also enjoyed reading this paper.

I have some clarification questions so as to ensure I did not miss something.

1. [Regarding the subtlety of privacy analysis of SELFSPAN]. If I understand it correctly, the problem is that even though the additional projection step does not touch sensitive data (labels) in this case, one cannot directly use post-processing. This is because the projection will leak the index of the sampling result, which then impacts the privacy amplification by subsampling in the original DP-SGD. Due to this, when using SELFSPAN, one cannot enjoy the gain/benefit of subsampling. On the other hand, since it does not touch labels, one can still use post-processing over the **non-subsampling** version of DP-SGD?

2. [Regarding the improvement over DP-SGD] It seems to me that the main gain over DP-SGD is the improvement over dimension d? It might be better to give more discussions on Table 4, as there are different choices of \sigma. Being more specific or using more particular values will be better I think.

3. [Confusion about the section name] For Section 4, the authors use SCO, which gives readers the impression that the goal is somehow the population excess loss. However, it turns out the authors are actually analyzing ERM. It might be better to replace the section name.

---

> ### Author Response · Authors · 2023-11-17
>
> We appreciate your insightful discussion. We were encouraged that you found our method “cute”, and we achieved “A good balance between theory and experimental results, which is a good fit for ICLR”. Here we address your detailed comments, which are helping us revise the paper and chart out future directions.
>
> **Q1. Theoretical contribution**
> > I think the comparisons in the theoretical part need to be clear, i.e., what's the exact gain of the proposed method?
>
> **A**: We thank the reviewer for this question. The goal of the theoretical section is to study the theoretical convex setting to build intuition for the general case of deep neural networks (which are beyond the reach of current theoretical frameworks). The goal is to provide some intuition for how LabelDP-Pro with different denoisers might compare to each other; in fact, the analysis of this setting is precisely what guided us to try the different denoisers that we do, but this section is by no means supposed to a complete explanation for the empirical results. Note that we provide the following caveat in the last paragraph of Section 4, _“Note that the bounds in Table 4 hold only for convex loss functions. Moreover, they are upper bounds on the excess error. For instance, while the bound in Lemma 3 is tight in the worst case, better rates might be achievable under additional assumptions on the loss function. Despite these limitations, we see in Table 4 that the excess error bounds align well with the experimental findings.”_
>
> **Q2. Non-applicability of Subsampling for SelfSpan**
> > If I understand it correctly, the problem is that even though the additional projection step does not touch sensitive data (labels) in this case, one cannot directly use post-processing. This is because the projection will leak the index of the sampling result, which then impacts the privacy amplification by subsampling in the original DP-SGD. Due to this, when using SELFSPAN, one cannot enjoy the gain/benefit of subsampling. On the other hand, since it does not touch labels, one can still use post-processing over the non-subsampling version of DP-SGD?
>
> **A**: Yes, indeed this is correct.
>
> Label-DP Pro using Self-Span projection can be viewed as a post-processing of non-subsampling version of DP-SGD, whereas Label-DP Pro using Alt-Span denoiser can be viewed as a post-processing of the subsampled version of DP-SGD.
> Perhaps a simple way to see this is as follows. Consider the following two mechanisms:
> 1. Mechanism $M_1$ that reveals, for each training step, just the noisy average gradient for the batch $I^G$ (without revealing $I^G$). Amplification by subsampling applies in this case.
> 2. Mechanism $M_2$ that reveals, for each training step, the entire _unlabeled_ batch $I^G$ along with noisy average gradients for that batch. Amplification by subsampling does not apply in this case.
>
> DP-SGD and LabelDP-Pro with Alt-Conv denoiser can be viewed as a post-processing of $M_1$, since in the latter, the projection operation only uses non-private information, namely the alt-batch $I^P$.
>
> On the other hand, LabelDP-Pro with Self-Span / Self-Conv denoisers cannot be seen as a post-processing of $M_1$, but can still be seen as a post-processing of $M_2$.

---

> > ### Author Response · Authors · 2023-11-17
> >
> > **Q3. Improvement over DP-SGD**
> > > It seems to me that the main gain over DP-SGD is the improvement over dimension d? It might be better to give more discussions on Table 4, as there are different choices of \sigma. Being more specific or using more particular values will be better I think.
> >
> > **A**: We firstly note that there is a small typo in the bound for `AltConv`, namely, the $\sigma$ should be $\sigma_a$, which is the same $\sigma_a$ present in the case of DP-SGD (`NoOp`). We have fixed this typo in the updated submission.The $\sigma_{na}$ in the case of `SelfSpan` and `SelfConv` are the same, and larger than $\sigma_a$ by roughly a factor of $\sqrt{n / n_1}$.
> >
> > If we compare the rows for `NoOp` and `AltConv`, we have indeed get rid of the dependence on $\sqrt{d}$ and replaced it by $\log^{0.25}(K n_2)$, which can be significantly smaller, although the denominator now is also a bit smaller ($\sqrt{T n_1}$ vs $\sqrt[4]{n_1}$). Thus, it is immediately clear that for a sufficiently large $d$, `AltConv` provides a better bound for any setting for parameters. We provide a more detailed comparison in Appendix E. As mentioned before we chose to keep the comparison qualitative, since as is common across deep learning, we use the theoretical convex setting to build intuition for the general case of deep neural networks (which are beyond the reach of current theoretical frameworks).
> >
> > We can perform a rough estimation of the quantities involved as follows.
> > Firstly, note that $\sigma_a$ is to be chosen such that $T$ steps of DP-SGD with batch size $n_1$ (that is, sub-sampling probability of $n_1 / n$) is roughly given as $\sigma_a \approx (n_1 / n) \cdot \sqrt{T} / \varepsilon \cdot \log(T / \delta)$ (using the advanced composition theorem, and [amplification by subsampling](https://arxiv.org/abs/1807.01647)).
> >
> > Suppose for concreteness, we chose $n_1 = n_2 = T = \sqrt{n}$ and using the bounds for $\sigma_a$ as above, we get the bounds in Table 4 (ignoring the $RC$ term) as
> > * for `NoOp` : $\sqrt{d} \cdot \frac{\log(n/\delta)}{\varepsilon n^{0.75}}$
> > * for `AltConv` : $\frac{\log^{0.25}(K n) \sqrt{\log(n/\delta)}}{n^{0.25}\sqrt{\varepsilon}}$
> >
> > And thus, the bound for `AltConv` is better when $d \gtrsim \frac{\varepsilon n \sqrt{\log (K n)}}{\log(n/\delta)}$. Note that there is nothing special about the particular setting of $n_1, n_2, T$ here. For any setting of these parameters, there is a large enough $d$ for which the error bound for `AltConv` will be better, since it does not depend on $d$ at all.
> >
> > **Q4. Section 4’s name**
> > > For Section 4, the authors use SCO, which gives readers the impression that the goal is somehow the population excess loss. However, it turns out the authors are actually analyzing ERM. It might be better to replace the section name.
> >
> > **A**: We thank the reviewer for raising this point. We consider ERM only for simplicity so that we do not need to add additional assumptions (involving more parameters) about the inherent stochasticity in the gradients. We’ve renamed the section to reflect that we are mainly analyzing Convex ERM.

---

> ### Author Response · Authors · 2023-11-21
> **Follow-up**
>
> We would like to extend our gratitude to the reviewer once again for their constructive feedback.
>
> Did our responses resolve the reviewer's concerns regarding the clarity of our theoretical results? Given that this appears to be the reviewer’s primary concern, we would like to ensure that we address it before the rebuttal deadline (Nov 22nd). We are also open to continuing discussions and welcome any further comments.

---

### Official Review · Reviewer_zF53 · 2023-10-31

**Soundness:** 3 good
**Presentation:** 3 good
**Contribution:** 3 good
**Rating:** 5
**Confidence:** 4

**Summary:**

This paper studies the algorithm design for label differential privacy (label-DP), where the labels in the dataset are private and the features are public. Previous label-DP algorithms are in the framework of random label flipping. This paper proposes a novel algorithm which utilizes the framework of DP-SGD. By the empirical evaluation, the proposed algorithm has the advantage over previous algorithms in the high-privacy regime.

**Strengths:**

1. The clarity is great. The arguments are well-explained and the structure is good.
2. The proposed algorithm is novel. It is different from previous algorithms which are in the framework of label random flipping. It instead utilizes the DP-SGD framework.
3. It derives the theoretical utility-privacy trade-off for DP convex optimization and compares this trade-off among different versions of their algorithm.
4. The proposed methods and baselines are evaluated on both the image benchmark and the Criteo dataset.

**Weaknesses:**

Both the proposed method and baseline numbers in the experiment can be potentially better. As they might be underestimated, the current comparison could be inaccurate, which is the main evaluation to illustrate the advantage of the proposed algorithm.
- The pure DP-SGD can be much better, which can potentially bring benefits for the proposed algorithm. For example, De et al., 2022 empirically show that DP-SGD can achieve 56.8% accuracy on CIFAR10 when $\varepsilon=1.0$, while the number reported in the paper is only 43.5%.
- ALIBI's accuracy on CIFAR-10 is reported as 51.3 when $\varepsilon=1.0$ in Table 5 in the paper, while in the original paper (Malek et al. (2021)) ALIBI achieves 71% when $\varepsilon=1.0$.
- When leveraging self-supervised learning with LabelDP, PATE-FM doesn't utilize the SelfSL on CIFAR-10, which seems a little unfair. It is reasonable to at least utilize a pretrained feature extractor as an initialization instead of random initialization when training teachers in PATE-FM. In this way, PATE-FM leveraging SelfSL is expected to have better accuracy.

Moreover, DP-SGD with gradient projection actually has been investigated for a while [1, 2] and not introduced in the paper. Although I still agree with the novelty of the proposed method given that all previous labelDP algorithms are based on label flipping, it might be worthwhile to see the difference and comparison between the proposed method and the methods in the literature.

[1] Yu, Da, et al. "Do not Let Privacy Overbill Utility: Gradient Embedding Perturbation for Private Learning." International Conference on Learning Representations. 2020.

[2] Kairouz, Peter, et al. "Fast dimension independent private adagrad on publicly estimated subspaces." arXiv preprint arXiv:2008.06570 (2020).

**Questions:**

Please check the details in "Weaknesses" above.

---

> ### Author Response · Authors · 2023-11-17
>
> We appreciate the reviewer’s insightful feedback. We were encouraged that you found our method novel and our arguments well-explained. Here we address your detailed comments, which are helping us revise the paper and chart out future directions.
>
> **Q1. Concerns about the accuracies**
> > The pure DP-SGD can be much better, which can potentially bring benefits for the proposed algorithm. For example, De et al., 2022 empirically show that DP-SGD can achieve 56.8% accuracy on CIFAR10 when eps=1.0 , while the number reported in the paper is only 43.5%.
>
> > ALIBI's accuracy on CIFAR-10 is reported as 51.3 when eps=1.0 in Table 5 in the paper, while in the original paper (Malek et al. (2021)) ALIBI achieves 71% when eps=1.0.
>
> **A**: Thanks for the comments! It is worth noting that the differences in accuracy between our results and those reported in previous studies can be attributed to the choice of a simpler convolutional network (see Table 10.b in Appendix D) as opposed to the ResNet used in earlier work. This variance in backbone models was driven by efficiency considerations, given our large-scale evaluations across multiple settings and benchmarks.
>
> However, as our method builds upon DP-SGD and reduces the noise it injects; if the baseline DP-SGD accuracy improves, we shall be able to achieve better accuracy. For instance, as shown below, when we switch to a more complicated network (with 4 more Conv layers), DP-SGD, ALIBI and our method all achieve higher performance, while our method is still able to retain the improvement for small $\varepsilon$’s.
>
> | $\varepsilon$ | ALIBI | DP-SGD | LabelDP-Pro |
> |---|---|---|---|
> | 0.05 | 9.9 | 10.3 | **18.4** |
> | 0.1 | 9.9 | 28.9 | **31.5** |
> | 0.2 | 22.4 | 36.0 | **38.3** |
> | 0.5 | 46.3 | 48.3 | **50.1** |
> | 1.0 | **68.7** | 51.6 | 52.3 |
>
> **Q2. Comparison w/ PATE-FM**
> > When leveraging self-supervised learning with LabelDP, PATE-FM doesn't utilize the SelfSL on CIFAR-10, which seems a little unfair. It is reasonable to at least utilize a pretrained feature extractor as an initialization instead of random initialization when training teachers in PATE-FM. In this way, PATE-FM leveraging SelfSL is expected to have better accuracy.
>
> **A**: Thank you for raising this concern. We want to clarify that while PATE-FM does not incorporate Self-Supervised Learning (SelfSL), its Semi-Supervised Learning (SemiSL) pipeline achieves notably higher accuracy (95.5% non-private accuracy) compared to the SelfSL pipeline we used in our evaluation, which achieves 90.9% non-private accuracy. This observation suggests that their backbone model is, in fact, **stronger** than our model.
>
> It is important to note that even though PATE-FM utilizes a more robust backbone model, our LabelDP-Pro approach continues to outperform PATE-FM in the high-privacy regime (see Table 6, $\varepsilon \leq 0.5$).

---

> > ### Author Response · Authors · 2023-11-17
> >
> > **Q3. Comparison with previous work**
> > > Moreover, DP-SGD with gradient projection actually has been investigated for a while [1, 2] and not introduced in the paper. Although I still agree with the novelty of the proposed method given that all previous labelDP algorithms are based on label flipping, it might be worthwhile to see the difference and comparison between the proposed method and the methods in the literature.
> >
> > > [1] Yu, Da, et al. "Do not Let Privacy Overbill Utility: Gradient Embedding Perturbation for Private Learning." International Conference on Learning Representations. 2020.
> >
> > > [2] Kairouz, Peter, et al. "Fast dimension independent private adagrad on publicly estimated subspaces." arXiv preprint arXiv:2008.06570 (2020).
> >
> > **A**: Thanks for sharing the references. [1] proposes Gradient Embedding Perturbation (GEP), which first projects individual private gradients into a non-sensitive low-dimensional anchor subspace, and then perturbs the low-dimensional embedding and the residual gradient separately according to the privacy budget. However, we note that [1] requires a public dataset to compute their gradient subspace; while such data is possible for image classification tasks, it is typically **not available for recommendation tasks**, which is one of the major applications of Label DP.
> >
> > That being said, we still present the results of [1] in the Table below for the review’s reference. As shown, [1] achieves similar performance to our methods: our LabelDP-Pro is better for smaller eps while [1] is better for larger ones.
> >
> > | $\varepsilon$ | DPSGD | GEP [1] | LabelDP-Pro (Ours) |
> > |---|---|---|---|
> > | 0.05 | 10.1 | 14.8 | **16.0** |
> > | 0.1 | 24.6 | 25.9 | **27.0** |
> > | 0.2 | 29.8 | 29.9 | **30.8** |
> > | 0.5 | 36.9 | 38.7 | **40.2** |
> > | 1.0 | 43.5 | **46.2** | 44.8 |
> >
> > [2] introduces a projection step to maintain the trajectory of the descent algorithm within the gradients' subspace. While this theoretically suggests faster convergence rates, empirical studies are not presented in the original work, making it challenging to establish a direct comparison.
> >
> > For completeness, we’ve updated our submission to discuss both papers as well (Appendix A).

---

> > > ### Comment · Reviewer_zF53 · 2023-11-23
> > > **Official Comment by Reviewer zF53**
> > >
> > > Thank the authors for the response!
> > >
> > > **Concerns on the accuracy.** Thanks for the clarifications on the experiment set-up.
> > >
> > > **Comparison w/ PATE-FM.** The comparison between non-private semi-supervised learning and self-supervised learning techniques seems not fair, because the performance of non-private self-supervised learning can be fully developed, while semi-supervised learning in PATE-FM is only developed with noisy labels so the non-private performance can be not fully leveraged.
> > >
> > > **Comparison with previous work.** Thanks for the clarification and it addresses my concern. I think it is better to move this discussion to the main paper instead of the appendix.
> > >
> > > Because PATE-FM is the SOTA label-dp algorithm and the advantage of the proposed algorithm over PATE-FM is not clear, I decide to keep my score.

---

> ### Author Response · Authors · 2023-11-21
> **Follow-up**
>
> We would like to extend our gratitude to the reviewer once again for their constructive feedback.
>
> Did our responses resolve the reviewer's concerns regarding 1) the accuracy when using a stronger backbone model, 2) our comparison with PATE-FM, and 3) the discussion and comparison with previous work? Given that they appear to be the reviewer’s primary concern, we would like to ensure that we address them before the rebuttal deadline (Nov 22nd). We are also open to continuing discussions and welcome any further comments.

---

### Official Review · Reviewer_Taxi · 2023-10-31

**Soundness:** 3 good
**Presentation:** 3 good
**Contribution:** 3 good
**Rating:** 6
**Confidence:** 5

**Summary:**

Paper proposes to use DP-SGD (private features and labels) in a label-only DP setting. It then proposes a way to de-noise DPSGD noisy gradients by projecting them to a convex hull build out of gradients derived only from the features (calculating gradients per existing label class).

Paper then shows how this de-noising process can be made compatible with privacy amplification by subsampling that is fundamental to utility-privacy trade off in DPSGD.

**Strengths:**

Paper proposes a novel gradient denoising method in label-only private settings. The proposed algorithm outperforms SOTA in high privacy regime on small public datasets like MNIST and CIFAR-10.

**Weaknesses:**

The main issue with applied DP is that production use cases really crave performances close to non-private regimes. Label private settings are very important for ads and marketing use cases and accuracy is of utmost value in these settings. Developers will prefer a medium privacy regime with very close to non-private utility compared to a highly private regime with very low utility.
From the paper results, trends suggest that the proposed algorithm gets inferior to SOTA as epsilon starts to pass 1 which I think is the most important hurdle in using the proposed algorithm in practice.

**Questions:**

I would like to see results of the proposed algorithm for epsilons between 1 and 10 too (for a fairer comparison and decision making for interested users).

I would like to see an empirical epsilon study (like the one referenced by authors, e.g. Malek et al. 2021)

I would like to see studies on CIFAR-100 (along with comparison with SOTA)

---

> ### Author Response · Authors · 2023-11-17
>
> We appreciate your insightful comments. We were encouraged that you found our method novel and acknowledged our improvement over SOTA. Here we address your detailed comments, which are helping us revise the paper and chart out future directions.
>
> **Q1. Concerns about targeting high-privacy regime**
> > Developers will prefer a medium privacy regime with very close to non-private utility compared to a highly private regime with very low utility. From the paper results, trends suggest that the proposed algorithm gets inferior to SOTA as epsilon starts to pass 1 which I think is the most important hurdle in using the proposed algorithm in practice.
>
> **A**: We appreciate this comment. We have explained why we consider the high-privacy regime in our [general response](https://openreview.net/forum?id=JnYaF3vv3G&noteId=6bj5h7E7Kl). In short, this consideration results from the intrinsic definition of DP and practical requirements for stronger privacy guarantees, as well as the unsatisfactory performance of existing LabelDP baselines.
>
> Furthermore, we note that in user-level privacy applications (as detailed in Table 7), our approach continues to outperform the baseline for medium-level privacy with $\varepsilon=5$, which demonstrates the versatility of our method in delivering competitive utility even in scenarios where a moderate level of privacy is required.
>
> **Q2. Results for moderate level of privacy**
> > I would like to see results of the proposed algorithm for epsilons between 1 and 10 too (for a fairer comparison and decision making for interested users).
>
> **A**: Thank you for your suggestion. Please find below the results for $\varepsilon$ values ranging from 0.05 to 10, with the best accuracy for each $\varepsilon$ highlighted in **boldface**. However, it is important to note that our method primarily focuses on achieving high privacy (i.e., the low-$\varepsilon$ regime). As a result, the relatively lower performance in larger $\varepsilon$ values is expected and has been explained in the theoretical analysis presented in Section 4 of our submission.
>
> | $\varepsilon$ | RR | RR-Debiased | LP-2ST | ALIBI | DP-SGD | LabelDP-Pro (Ours) |
> |---|---|---|---|---|---|---|
> | 0.05 | 10.0 | 10.0 | 10.0 | 9.9 | 10.0 | **16.0** |
> | 0.1 | 10.0 | 10.0 | 10.0 | 9.9 | 24.6 | **27.0** |
> | 0.2 | 10.0 | 10.0 | 10.0 | 17.9 | 29.8 | **30.8** |
> | 0.5 | 10.0 | 10.0 | 35.8 | 34.4 | 36.9 | **40.2** |
> | 1.0 | 17.0 | 10.0 | **60.9** | 51.3 | 43.5 | 44.8 |
> | 2.0 | 77.6 | 58.8 | 63.8 | **64.2** | 51.4 | 52.0 |
> | 5.0 | 80.4 | 80.2 | 81.7 | **82.3** | 59.5 | 59.8 |
> | 10.0 | 81.5 | 81.8 | 83.5 | **83.5** | 63.6 | 63.7 |
>
> **Q3. Empirical epsilon study**
> > I would like to see an empirical epsilon study (like the one referenced by authors, e.g. Malek et al. 2021)
>
> **A**: Thank you for your suggestion. While we acknowledge the importance of privacy auditing and empirical privacy studies, it appears to be outside the scope of our current work. We intend to consider this for future research endeavors.
>
> **Q4. Results on CIFAR-100**
> > I would like to see studies on CIFAR-100 (along with a comparison with SOTA)
>
> **A**: We appreciate this suggestion and are working on gathering these results. We will try to share them in a later post.

---

> > ### Author Response · Authors · 2023-11-22
> >
> > **Q4. Results on CIFAR-100**
> >
> > **A**: Please find below the test accuracy on CIFAR-100, using the model in Table 10.b (in the appendix). Note that we are comparing to ALIBI note PATE-FM, as ALIBI achieves better results than PATE-FM according to Malek et al. 2021 (see their Table 1).  As shown, LabelDP-Pro is able to achieve better performance than RR for $\varepsilon \leq 5.0$.
> >
> > | $\varepsilon$ | RR | LP-MST | ALIBI | DP-SGD | LabelDP-Pro (Ours) |
> > |---|---|---|---|---|---|
> > | 0.1 | 1.0 | 1.1 | 0.9 | 1.1 | 1.1 |
> > | 0.2 | 1.0 | 1.1 | 1.0 | 1.1 | 1.1 |
> > | 0.5 | 1.0 | 1.1 | 1.4 | 1.7 | 1.8 |
> > | 1.0 | 1.0 | 1.4 | 2.8 | 9.4 | 9.7 |
> > | 2.0 | 1.1 | 3.5 | 14.4 | 27.4 | 27.7 |
> > | 5.0 | 33.5 | 35.6 | 40.7 | 46.8 | 47.1 |
> >
> > We would like to extend our gratitude to the reviewer once again for their constructive feedback. Did our response resolve the reviewer's concerns regarding the focus on high-privacy regime? Given that this appears to be the reviewer’s primary concern, we would like to ensure that we address it before the rebuttal deadline (Nov 22nd). We are also open to continuing discussions and welcome any further comments.

---

### Official Review · Reviewer_3Pz1 · 2023-11-01

**Soundness:** 3 good
**Presentation:** 3 good
**Contribution:** 3 good
**Rating:** 6
**Confidence:** 3

**Summary:**

This work considers training under label differential privacy by projecting the DP-SGD gradient onto a gradient subspace that only depends on the features. Experiments show that the algorithm works well in the high privacy regime ($\varepsilon <1$). Theoretical analysis proves convergence in the stochastic convex optimization setting.

**Strengths:**

1. The idea to project the noisy gradient of DP-SGD onto a smaller subspace is simple and intuitive.
2. The authors also address practical issues of memory efficiency and stability when calculating projections, which makes the algorithm more practical.
3. The experiment result is encouraging for the high privacy regime ($\varepsilon < 1$)
4. The algorithm has theoretical support which shows that dimension-independent convergence rate can be achieved under stochastic convex optimization.

**Weaknesses:**

1. The user-level private algorithm only leverages group privacy, which is extremely sub-optimal even in the simple problem of mean estimation. A good user-level private algorithm should go beyond the simple application of group privacy.
2. For $\varepsilon \ge 1$ in Tables 5 and 6, the proposed algorithm underperforms the best baseline by a large margin. In practice, it is still reasonable to use $\varepsilon < 5$, so it would be better if the algorithm could also perform well for moderate level of privacy.

**Questions:**

1. Could you explain why the proposed algorithm underperforms the best baseline by a large margin when training from scratch for $\varepsilon \ge 1$?
2. When using pre-trained weights from SSL, the result for $\varepsilon > 1$ is much better. Maybe this is due to a more accurate gradient subspace after SSL pretraining?

---

> ### Author Response · Authors · 2023-11-17
>
> We appreciate your insightful input. We were encouraged that you found our method simple yet practical, and our empirical results encouraging. Here we address your detailed comments, which are helping us revise the paper and chart out future directions.
>
> **Q1. Sub-optimal choice for group privacy**
> > The user-level private algorithm only leverages group privacy, which is extremely sub-optimal even in the simple problem of mean estimation. A good user-level private algorithm should go beyond the simple application of group privacy.
>
> **A**:  We appreciate this comment. We would like to note that we ensure a fair comparison among all methods by **employing the same group privacy rule for all approaches**. Furthermore, we evaluated our approach not only with user-level privacy (Section 6), we also showed that it consistently outperforms the baselines in item-level privacy settings (Section 5).
>
> We would also like to note that, while there are many recent results on user-level privacy that go beyond group privacy [1,2,3], none of the results are practical as far as we can tell (e.g. in the use of the i.i.d. or other assumptions on the data). This is demonstrated by the lack of empirical evaluations in these works. We would appreciate it if the reviewer could provide further clarification or pointers to prior work on practical user-level privacy that goes beyond simple group privacy.
>
> [1] Daniel Levy, Ziteng Sun, Kareem Amin, Satyen Kale, Alex Kulesza, Mehryar Mohri, Ananda Theertha Suresh. “Learning with User-Level Privacy.” https://arxiv.org/abs/2102.11845
>
> [2] Esfandiari, Hossein, Vahab Mirrokni, and Shyam Narayanan. "Tight and robust private mean estimation with few users." https://arxiv.org/abs/2110.11876
>
> [3] Badih Ghazi, Pritish Kamath, Ravi Kumar, Raghu Meka, Pasin Manurangsi, Chiyuan Zhang. “On User-Level Private Convex Optimization.” https://arxiv.org/abs/2305.04912
>
> **Q2. Performance for moderate level of privacy**
> > For eps>=1 in Tables 5 and 6, the proposed algorithm underperforms the best baseline by a large margin. In practice, it is still reasonable to use eps<5, so it would be better if the algorithm could also perform well for moderate level of privacy.
>
> **A**: We appreciate this comment. We’ve explained why we consider the high-privacy regime in our [general response](https://openreview.net/forum?id=JnYaF3vv3G&noteId=6bj5h7E7Kl). In short, this consideration results from the intrinsic definition of DP and practical requirements for stronger privacy guarantees, as well as the unsatisfactory performance of existing LabelDP baselines.
>
> Furthermore, we note that in user-level privacy applications (as detailed in Table 7), our approach continues to outperform the baseline for medium-level privacy with $\varepsilon=5$, which demonstrates the versatility of our method in delivering competitive utility even in scenarios where a moderate level of privacy is required.
>
> **Q3. Explanation for inferior performance for eps>=1**
> > Could you explain why the proposed algorithm underperforms the best baseline by a large margin when training from scratch for eps>=1
>
> **A**: Thanks for this question. A possible explanation for this is that the excess variance in the gradients introduced by randomized response decreases exponentially as ε increases beyond 1. On the other hand, in DP-SGD, the excess variance in the gradients decreases at best only polynomially $1/\varepsilon^{O(1)}$. This is analyzed formally in the case of convex optimization in Table 4 of our submission. This distinction can provide an explanation of why our method, which is built on DP-SGD, outperforms baseline methods utilizing randomized response in the low-ε regime but exhibits inferior performance in the high-ε regime.
>
> **Q4. SSL outperforms trained from scratch**
> > When using pre-trained weights from SSL, the result for eps>1  is much better. Maybe this is due to a more accurate gradient subspace after SSL pretraining?
>
> **A**: Thank you for this question. Yes, the accuracies achieved under SSL (Table 6) are higher than those achieved when trained from scratch (Table 5), which could be attributed to better image representations learnt by SSL and thus resulting in a more accurate gradient subspace.

---

> ### Comment · Reviewer_3Pz1 · 2023-11-18
> **Thanks for your comment.**
>
> ### **Results for moderate and large epsilon**
> Thanks for your clarification and explanation. I think the large gap between the proposed method and existing methods in the medium to low privacy regimes is still my biggest concern. If the method is only marginally inferior to the baselines for $\varepsilon>1$, I would be much more supportive of this work as it proposes a new projective gradient method for Label DP. My opinion remains unchanged for now.
>
> ### **Comments on user-level privacy**
> I do not quite agree with the argument that  "practical applications of user-level DP cannot go beyond group privacy". Many works in federated learning have integrated user-level privacy. See Figure 2 in [1] and a more recent work of [2]. I don't think any of these works use a simple application of group privacy. These works apply minibatch gradients in DP-SGD/ DP-Fed-Avg for each user, replacing per-example gradients in standard item-level DP-SGD, which should naturally provide user-level privacy guarantees. Using a per-batch gradient would help reduce the variance, which is precisely the important intuition that was used in the three theoretical works you mentioned.
>
> Moreover, the experiments you have demonstrated for user-level DP have no more than 10 samples per user, which is far from practical. Even with such a small $k$, we can also see as $k$ increases, the performance actually drops, which is another empirical evidence that group privacy is highly sub-optimal.  I am curious about what happens when k is slightly larger, say 100, which is very common in practical scenarios. I view this part of the result more as a demonstration of good performance for very small epsilon, rather than an independent and novel contribution to user-level differential privacy.
>
> [1] Practical and Private (Deep) Learning Without Sampling or Shuffling, Kairouz et.al.  ICML 2021
> [2] Learning to Generate Image Embeddings with User-level Differential Privacy, Xu et.al, CVPR 2023

---

> ### Author Response · Authors · 2023-11-22
>
> **1. On high v.s. low privacy**
> > I think the large gap between the proposed method and existing methods in the medium to low privacy regimes is still my biggest concern. If the method is only marginally inferior to the baselines for $\varepsilon>1$, I would be much more supportive of this work as it proposes a new projective gradient method for Label DP.
>
> **A**: Our method serves as a better LabelDP option in high-privacy regimes. However, it's worth noting that developers retain the flexibility to choose SOTA methods for low-privacy regimes. Furthermore, we have conducted an empirical study on the point of intersection (as detailed in Appendix D.4), which can serve as a reference for making decisions about which method to deploy.
>
> Furthermore, we wish to point out that it is common for different DP algorithms to be better in different regimes of parameters. For example, for linear queries, there are three pure-DP algorithms which are better for different regimes of parameters; see Theorem 1 in [A].
>
> References:
> [A] Aleksandar Nikolov: Private Query Release via the Johnson-Lindenstrauss Transform. SODA 2023: pp. 4982-5002.
>
>
> **2. On user-level privacy**
> > I am curious about what happens when k is slightly larger, say 100, which is very common in practical scenarios.
>
> **A**: Below we present the results for $k=100$. As shown, our method consistently outperforms RR and DP-SGD under $k=100$.
>
> | $\varepsilon$ | RR | DP-SGD | LabelDP-Pro |
> |---|---|---|---|
> | 0.1 | 0.216 | 0.634 | 0.675 |
> | 0.2 | 0.218 | 0.662 | 0.694 |
> | 0.5 | 0.251 | 0.697 | 0.731 |
> | 1.0 | 0.577 | 0.701 | 0.736 |
> | 2.0 | 0.643 | 0.709 | 0.743 |
> | 5.0 | 0.698 | 0.726 | 0.748 |
>
> > I do not quite agree with the argument that "practical applications of user-level DP cannot go beyond group privacy". Many works in federated learning have integrated user-level privacy. See Figure 2 in [1] and a more recent work of [2]. I don't think any of these works use a simple application of group privacy. These works apply minibatch gradients in DP-SGD/ DP-Fed-Avg for each user, replacing per-example gradients in standard item-level DP-SGD, which should naturally provide user-level privacy guarantees.
>
> **A**: We appreciate the reviewer’s prompt response and for providing the references. We would like to clarify that what the reviewer suggested is exactly our implementation for user-level DP-SGD and LabelDP-Pro. We use group privacy only for RR-based methods. We apologize for any lack of clarity in our initial submission, and for the confusion in our previous response.
>
> We agree with the reviewer and have observed that opting for group privacy results in a decrease in performance for these two gradients-involved methods. This indeed underscores another advantage of our approach, as it is compatible with a more effective user-level privacy implementation.
>
> We have updated our submission to reflect these details more explicitly (Section 6) and have also included the provided references.

---

### Author Response · Authors · 2023-11-17
**General Response**

We thank the AC and all the reviewers for their time and valuable feedback, and for the recognition that our LabelDP-Pro method is "simple and effective", and our submission shows “a good balance between theory and experimental results".

**1. Why we target high-privacy regimes**

Our emphasis on high-privacy regimes stems from several important practical reasons:
- **Usefulness of high-privacy**: high privacy is especially useful in DP for two reasons: (i) composition - if a bigger system consists of multiple components, it is essential to ensure each component maintains high privacy to avoid low-composed privacy;  and (ii) user-level, which is the most desirable guarantee - regardless of the group privacy composition, the item-level $\varepsilon$ has to be sufficiently small for user-level to have a meaningful $\varepsilon$.
- **Real-world requirements of high-privacy application**: Developers' privacy requirements can greatly vary based on factors such as the specific use case, industry regulations, or the nature of the application. In certain contexts, particularly those involving highly sensitive data, developers may prioritize heightened privacy levels, even if it entails some reduction in utility. For instance, in the table below, we provide examples of applications by major tech companies that have chosen to prioritize very high privacy levels, which confirms the importance of the high-privacy regime for practical deployments.
- **Addressing the shortcomings of existing baselines**: Existing baseline LabelDP methods (e.g., based on label randomization) often struggle to perform adequately in high-privacy regimes and resulting in nearly random outcomes. Our proposal represents a significant step towards making LabelDP a practical and viable choice even in high-privacy scenarios, thereby addressing a pressing need in the field.

| Company | Privacy Model | Application | $\varepsilon$ |
|---|---|---|---|
| Google | Local | [Google - RAPPOR Chrome Homepages]( https://static.googleusercontent.com/media/research.google.com/en//pubs/archive/42852.pdf) | 0.534  |
| Microsoft | Local | [Telemetry Collection per App]( https://arxiv.org/abs/1712.01524) | 0.686 |
| Apple | Local | [Learning with Privacy at Scale (Health Type Usage)](https://docs-assets.developer.apple.com/ml-research/papers/learning-with-privacy-at-scale.pdf) | 2.0 |
| Meta (Facebook) | Global | [Facebook Privacy-Protected Full URLs Data Set](https://dataverse.harvard.edu/dataset.xhtml?persistentId=doi:10.7910/DVN/TDOAPG) | 1.45 |
| LinkedIn | Global | [LinkedIn's Audience Engagements API](https://arxiv.org/abs/2002.05839) | 0.15 |

**2. Updates to the submission**

Based on the reviewers’ feedback, we’ve made the following changes to our submission:
* Updated the theoretical analysis (Reviewer NJnW): we’ve updated Section 4’s title and fixed a typo.
* Added results table for larger $\varepsilon$'s (Reviewer Taxi): we’ve added results for larger $\varepsilon$’s in Table 13 (Appendix D.4).
* Discussed related work (Reviewer zF53): we’ve also discussed previous proposals that have explored DP-SGD with gradient project (Appendix A).

---

### Meta-Review · Area_Chair_vvVc · 2023-12-04

**Metareview:**

The paper presents a method for label-DP SGD by projecting the noisy gradient to a subspace or polytope computed from gradients of public inputs with all possible labels. The method improves accuracy over previous state-of-the-art especially in strong privacy regime.

I read the paper myself in detail and was impressed by its quality: covering different variants of the method correctly.

In general, I do not agree with the criticism raised by some reviewers that strong privacy regime is not important. The strong privacy regime is exactly what we would want, especially in label DP where the only thing being protected is a single label and the protection from state-of-the-art randomized response with even $\epsilon=2$ is quite weak.

Strengths: carefully presented method covering all aspects of the work. Addressing the strong privacy regime is important.

Weaknesses: improves over state-of-the-art only in strong privacy regime where the benchmark results are too poor to be practical for the models/problems studied in the paper.

**Justification For Why Not Higher Score:**

While the method is interesting, the authors do not demonstrate examples where it would improve over previous work in a regime with good enough accuracy to be practical.

**Justification For Why Not Lower Score:**

This is a solid paper improving over previous state-of-the-art in a relevant problem.

---

### Decision · Program_Chairs · 2024-01-16

Accept (poster)